# Asymptotic Properties and Application of GSB Process: A Case Study of the COVID-19 Dynamics in Serbia

**Mihailo Jovanović** [1,†]**, Vladica Stojanović** [2,*,†] **, Kristijan Kuk** [2,†]**, Brankica Popović** [2,†] **and Petar Čisar** [2,†]

1    The Office for Information Technologies and eGovernment, 11000 Belgrade, Serbia
2    Department of Informatics & Computer Sciences, University of Criminal Investigation and Police Studies, 11000 Belgrade, Serbia
*    Correspondence: vladica.stojanovic@kpu.edu.rs
†    These authors contributed equally to this work.

**Abstract:** This paper describes one of the non-linear (and non-stationary) stochastic models, the GSB (Gaussian, or Generalized, Split-BREAK) process, which is used in the analysis of time series with pronounced and accentuated fluctuations. In the beginning, the stochastic structure of the GSB process and its important distributional and asymptotic properties are given. To that end, a method based on characteristic functions (CFs) was used. Various procedures for the estimation of model parameters, asymptotic properties, and numerical simulations of the obtained estimators are also investigated. Finally, as an illustration of the practical application of the GSB process, an analysis is presented of the dynamics and stochastic distribution of the infected and immunized population in relation to the disease COVID-19 in the territory of the Republic of Serbia.

**Keywords:** stochastic processes; emphatic fluctuations; non-stationarity; asymptotic normality; Gaussian distribution; estimation; COVID-19

**MSC:** 60E10; 60F05; 62M10





## 1. Introduction

Stochastic models which are used in the analysis of time series with pronounced and permanent fluctuations are of particular importance in contemporary research. For this purpose, we start from the basic results of Engle and Smith [1], who first introduced the so-called STOchastic Permanent BREAKing process, popularly called the *STOPBREAK process*. Many authors have since considered the STOPBREAK notion, primarily in the field of econometrics. Some of its modifications were considered, among others, in [2–5], while its application was presented, for instance, in [6–8].

The original modification of the STOPBREAK process, named *the Split-BREAK model*, was introduced in [9]. After that, the general form of this process, named *Gaussian (or Generalized) Split-BREAK (GSB) process*, was proposed in [10–12]. This stochastic model also can be viewed as a generalization of STOPBREAK, as well as a well-known linear Auto-Regressive Moving Average (ARMA) model. In that way, the GSB process has already been applied in analyzing non-linear time series with pronounced and permanent fluctuations. Let us point out that in the mentioned works, of main consideration were the stochastic properties of the stationary components of the GSB process. The main goal of this paper is a more detailed investigation of the non-stationary components (time series) of the GSB model. These series naturally have a more complex stochastic structure, but they are of particular interest in contemporary research [13–18]. To this end, the asymptotic properties of distributions of the GSB series will also be of specific interest.

In addition to the theoretical aspects, the application of the GSB process in describing the dynamics and finding an adequate stochastic distribution of the infected and immunized population with respect to COVID-19 on the territory of the Republic of Serbia was also

considered. We point out that many authors who deal with this, still current, issue have contributed various theoretical models that investigate it from several aspects. For instance, rigorous mathematical models, usually based on analyzing and solving systems of partial coupled equations, have been proposed, among others, in [19–21]. On the other hand, works in [22–25] combine deterministic and stochastic approaches, such as multiple and logistic regression, multifactor correlation, and the least squares estimation method, to predict the various effects caused by the COVID-19 pandemic. A particularly interesting approach is given in [26,27] where, to predict the COVID-19 dynamics more accurately, machine learning techniques and the construction of a complete information system are used. Finally, to the best of our knowledge, most stochastic approaches to-date in the analysis of infection, immunization, and other indicators related to the disease of COVID-19 were based on the use of the gamma distribution [21,28], as well as a log-normal distribution [29]. This is precisely one of the reasons why we believe that a different approach is given here, primarily in stochastic modeling and research of this problem. At the same time, let us emphasize that our main goal is to model the temporal dynamics of the COVID-19 disease, based on a formal study of the stochastic structure of the GSB model. In this sense, some other indicators and features of this disease, which can also affect its dynamics (see, for instance [30–32]), can to a certain degree be a limitation of this approach.

In the next section, starting from previous works [9–12], some definitions and basic stochastic properties of the GSB process are discussed. Section 3 contains the main and novel results related to this process's detailed stochastic structure and asymptotic properties, where the method of characteristic functions (CFs) was used as the basic tool. Section 4 presents the procedure for estimating the unknown parameters of the GSB process and an investigation of the asymptotic properties of the obtained estimators. Numerical Monte Carlo simulations of the obtained estimators are considered in Section 5. In addition, the application of the GSB process in describing the dynamics and distribution of the size of infected and immunized populations on the territory of the Republic of Serbia is given here. Finally, concluding remarks are highlighted in Section 6.

## 2. Definition and Main Properties of the GSB Process

The basic series of GSB processes is defined by the following equality:

$$y_t = m_t + \varepsilon_t. \tag{1}$$

Here, $t = 0, 1, \ldots, T$ are the known time values, $(m_t)$ is the series of the so-called *martingale means*, and $(\varepsilon_t)$ are *the innovations*, i.e., series of independent identical distributed (IID) Gaussian $\mathcal{N}(0, \sigma^2)$ random variables (RVs). Moreover, it is considered that $(\varepsilon_t)$ is defined on the same probability space $(\Omega, \mathcal{F}, P)$, expanded by some filtration $F = (\mathcal{F}_t)$, i.e., nondecreasing $\sigma$-algebras on $\Omega$. In a practical sense, filtration $(\mathcal{F}_t)$ represents a set of "information" at time $t$. Therefore, it is assumed that, for each $t = 0, 1, \ldots, T$, the RVs $\varepsilon_t$ are $\mathcal{F}_t$-adaptive. Accordingly, the conditional expectation, as well as the variance of RVs $\varepsilon_t$, are, respectively,

$$E(\varepsilon_t | \mathcal{F}_{t-1}) = 0, \quad V(\varepsilon_t | \mathcal{F}_{t-1}) = E\left(\varepsilon_t^2 \middle| \mathcal{F}_{t-1}\right) = \sigma^2.$$

On the other hand, for martingale means $(m_t)$, we assume that they are defined by the following recurrence relation:

$$m_t = m_{t-1} + q_{t-1}\varepsilon_{t-1} = m_0 + \sum_{j=0}^{t-1} q_j \varepsilon_j. \tag{2}$$

Here, we can effectively assume that $m_0 \overset{as}{=} \mu$ (*const.*) and $\varepsilon_{-1} = \varepsilon_0 \overset{as}{=} 0$. Meanwhile, $q_t$ is the so-called *noise indicator*, i.e., the RV that depends on innovations ($\varepsilon_t$) in the following way:

$$q_t = I\left(\varepsilon_{t-1}^2 > c\right) = \begin{cases} 1, & \varepsilon_{t-1}^2 > c \\ 0, & \varepsilon_{t-1}^2 \leq c. \end{cases}$$

The value $c > 0$ represents *the critical value of the reaction*, i.e., the significance of the previous realization of innovations ($\varepsilon_t$) which allow their present values to be included in Equation (2). In other words, value $q_{t-1} = 0$ indicates that there is no change in the martingale mean value $m_t$, compared to the previous value $m_{t-1}$. Consequently, the value $y_t$ will be obtained with a "small" fluctuation, which depends only on $\varepsilon_t$. By contrast, in the case of $q_t = 1$ an emphatic (permanent) fluctuation of $y_t$ is registered. Thus, the level of previous realizations of series ($\varepsilon_t$) affects the degree of variations in the series ($y_t$), that is, it indicates the intensity of fluctuations in the GSB process. Furthermore, according to the previous equalities, it follows that:

$$E(y_t|\mathcal{F}_{t-1}) = m_t + E(\varepsilon_t|\mathcal{F}_{t-1}) = m_t,$$

from which we conclude that the series realizations ($y_t$) are "close" to the martingale means ($m_t$). Moreover, it is valid to put:

$$\begin{aligned} E(y_t) & = E[E(y_t|\mathcal{F}_{t-1})] = E(m_t) = E(m_{t-1}) + E(q_{t-1}\varepsilon_{t-1}) \\ & = E(m_{t-1}) = \cdots = E(m_0) = \mu, \end{aligned}$$

i.e., the mean values of the series ($y_t$) and ($m_t$) have equal, constant values. We notice that the previous equalities speak a lot about the stochastic nature of the GSB process, that is, the additive decomposition (1). Since the sequence ($m_t$) is measurable concerning the field $\mathcal{F}_{t-1}$, it represents a component of *predictability and stability* of the GSB process. In contrast, the innovations series ($\varepsilon_t$) is *the deviation factor (white noise)* of the basic GSB series ($y_t$) in relation to the martingale means ($m_t$).

Further, we determine the conditional variance of the series ($y_t$) from the equation:

$$V(y_t|\mathcal{F}_{t-1}) = E(y_t^2|\mathcal{F}_{t-1}) - m_t^2 = 2m_t E(\varepsilon_t) + E(\varepsilon_t^2) = \sigma^2,$$

and from here, one obtains:

$$V(y_t) = E(y_t^2) - \mu^2 = E(m_t^2) + 2E(m_t\varepsilon_t) + E(\varepsilon_t^2) - \mu^2 = V(m_t) + \sigma^2.$$

For each $t = 1, \ldots, T$, it also holds that:

$$\begin{aligned} V(m_t) & = E\left(m_t^2\right) - \mu^2 \\ & = E\left(m_{t-1}^2\right) + 2E(m_{t-1}q_{t-1}\varepsilon_{t-1}) + E\left(q_{t-1}^2\varepsilon_{t-1}^2\right) - \mu^2 \\ & = V(m_{t-1}) + a_c\sigma^2, \end{aligned}$$

where $a_c = E(q_t) = E\left(q_t^2\right) = P\{\varepsilon_t^2 > c\}$. It follows that the variance of martingale means ($m_t$), under the assumption $m_0 \equiv \mu$ (*const.*), can be expressed as:

$$V(m_t) = t a_c \sigma^2, t \geq 0.$$

From here, the variance of the basic series ($y_t$) can be obtained as follows:

$$V(y_t) = V(m_t) + \sigma^2 = (t a_c + 1)\sigma^2, \ t \geq 0.$$

According to the previous equalities, the variances of the series ($y_t$) and ($m_t$) have non-constant values that depend on the point in time ($t$) in which they are observed.

Correlation functions of the series $(y_t)$ and $(m_t)$ can be obtained in a similar way. Note that for every $s > t \geq 0$, it holds that:

$$
\begin{aligned}
Cov(m_t, m_s) &= E(m_t m_s) - \mu^2 = E(m_t m_{s-1}) + E(m_t q_{s-1} \varepsilon_{s-1}) - \mu^2 \\
&= Cov(m_t, m_{s-1}),
\end{aligned}
$$

and it is easy to see that the covariance of the series $(m_t)$ satisfies:

$$
Cov(m_t, m_s) = V(m_t), \quad s > t \geq 0.
$$

From here, the correlation function of the martingale means is obtained:

$$
\widetilde{K}(s, t) = \frac{Cov(m_t, m_s)}{\sqrt{V(m_t)} \cdot \sqrt{V(m_s)}} = \begin{cases} \frac{\min(s,t)}{\sqrt{s \cdot t}}, & s \neq t \\ 1, & s = t. \end{cases}
$$

Similarly, according to equalities:

$$
\begin{aligned}
Cov(y_t, y_s) &= E(y_t y_s) - \mu^2 = E(y_t m_s) + E(y_t \varepsilon_s) - \mu^2 \\
&= E(m_t m_s) + E(\varepsilon_t m_s) - \mu^2 = Cov(m_t, m_s) + a_c \sigma^2 \\
&= V(m_t) + a_c \sigma^2 = V(y_t), \quad s > t \geq 0,
\end{aligned}
$$

the correlation function for $(y_t)$, can be obtained as follows:

$$
K(s, t) = \begin{cases} \frac{a_c \min(s,t) + 1}{\sqrt{(a_c s + 1) \cdot (a_c t + 1)}}, & s \neq t \\ 1, & s = t. \end{cases}
$$

Therefore, both correlation functions depend on the time arguments $t, s$ and indicate the non-stationarity of the series $(y_t)$ and $(m_t)$. This fact requires some more complex techniques to examine their properties. Moreover, note that when $s > t \geq 0$,

$$
\begin{aligned}
\lim_{s \to t} \widetilde{K}(s, t) &= \lim_{s \to t} \frac{\min(s,t)}{\sqrt{s \cdot t}} = \frac{t}{\sqrt{t^2}} = 1 \\
\lim_{s \to t} K(s, t) &= \lim_{s \to t} \frac{a_c \min(s,t) + 1}{\sqrt{(a_c s + 1) \cdot (a_c t + 1)}} = \frac{a_c t + 1}{\sqrt{(a_c t + 1)^2}} = 1.
\end{aligned}
$$

Thus, the correlation functions of both series $(y_t)$ and $(m_t)$ satisfy the $L^2$-continuity condition.

At the end of this section, we define *a series of increments of the GSB process* by the following equality:

$$
X_t = y_t - y_{t-1}, \quad t = 1, \ldots, T. \tag{3}
$$

Almost all authors who have studied STOPBREAK processes highlight the importance of this sequence. This series, as can be easily seen from Equations (1) and (2), can be given in the following form:

$$
X_t = \varepsilon_t - \theta_{t-1} \varepsilon_{t-1}, \tag{4}
$$

where $\theta_t = 1 - q_t = I(\varepsilon_{t-1}^2 \leq c)$. The series $(X_t)$ is named *a Splitting Moving Average process* (*of order* 1), shortened to *Split-MA (1) process*, because it operates in two regimes. Fluctuations of innovations $(\varepsilon_t)$ that were emphasized in the previous time moment $(t-1)$ imply $\theta_{t-1} = 0$, so the equality $X_t = \varepsilon_t$ holds. On the other hand, fluctuations that do not exceed the critical value $c$ give a representation of $(X_t)$ in the form of a standard, linear MA (1) process. In this way, $(X_t)$ has similar properties to the MA (1) models, which can be applied in research into it. Thus, taking earlier assumptions, the mean value and variance of this series, obtained by simple computation, are:

$$
E(X_t) = 0, \quad V(X_t) = E\left(X_t^2\right) = \sigma^2(b_c + 1),
$$

where $b_c = 1 - a_c = P(\varepsilon_{t-1}^2 \leq c)$. Moreover, the covariance of this sequence is:

$$Cov(X_t, X_s) = \begin{cases} (b_c + 1)\sigma^2, & s = t \\ -b_c\sigma^2, & |s - t| = 1 \\ 0, & \text{otherwise,} \end{cases}$$

and obviously has an identical structure to the standard MA (1) series. Based on the obtained covariance, we can easily see that the series $(X_t)$ is stationary and that its correlation function can be written in the form:

$$\rho_X(h) := \frac{Cov(X_t, X_{t+h})}{V(X_t)} = \begin{cases} 1, & h = 0 \\ -b_c/(b_c + 1), & h = \pm 1 \\ 0, & \text{otherwise.} \end{cases}$$

Finally, according to Equations (3) and (4), it follows that:

$$y_t - y_{t-1} = \varepsilon_t - \theta_{t-1}\varepsilon_{t-1}, \ t = 1, \ldots, T,.$$

which can be viewed as a non-linear *Integrated Auto-Regressive Moving Average (ARIMA) model* with "temporary" components $(\theta_{t-1}\varepsilon_{t-1})$. These imply the specific structure of the series $(X_t)$, as well as other components of the GSB process.

In the following section, as we have already pointed out, we also discuss the application of the GSB model in describing the dynamics of infection and immunization of the population on the territory of the Republic of Serbia. As will be seen, this kind of dynamics has pronounced fluctuations that can be described by the non-stationary components of the GSB process, primarily by its main time series $(y_t)$. In that case, due to its stationarity, the Split-MA (1) process plays an important role. As an illustration, Figure 1 shows the realizations of all the above-mentioned series obtained by the Monte Carlo simulation of the GSB model.

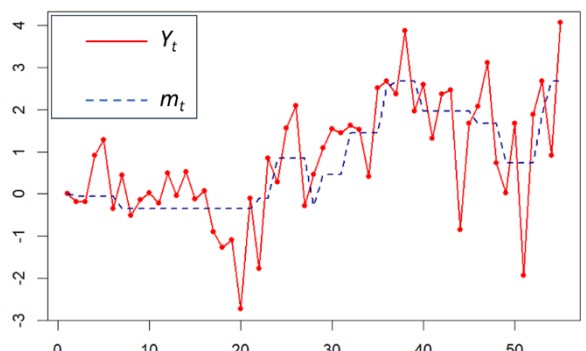 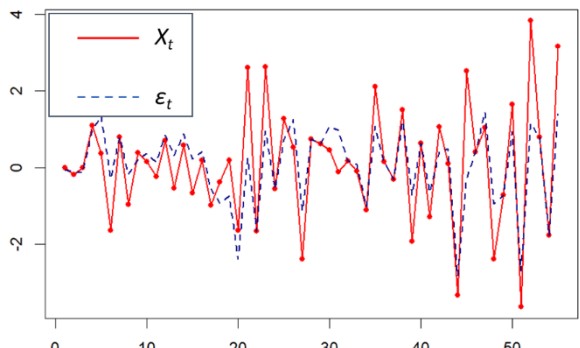

**Figure 1.** Dynamics of the basic series of the GSB model. (Parameter values are: $\mu = 0$ and $c = \sigma = 1$).

## 3. Stochastic Distribution and Asymptotic Properties of the GSB Process

In this section, some stochastic properties of the GSB process, regarding the distribution and asymptotic behavior of its basic stochastic components, are discussed in more detail. As explained in the previous section, the GSB model, given by Equations (1)–(4), contains four stochastic components: the basic series $(y)$, innovations $(\varepsilon_t)$, the martingale means $(m_t)$, and the series of increments $(X_t)$. At the same time, series $(\varepsilon_t)$ and $(X_t)$ represent the stationary components of the GSB process, where $(X_t)$ is "close" to the linear MA model. In general form, the stochastic structure of the series $(X_t)$ is described in [12], where the method of characteristic functions (CFs) was used. Following this approach, the basic stochastic properties of the series $(X_t)$ can be expressed by the following statement.

**Theorem 1.** *Let $(X_t)$ be the Split-MA (1) process defined by Equation (4). For arbitrary $x \in \mathbb{R}$ and $t = 0, \cdot 1, \ldots, T$, the cumulative distribution function (CDF) of this stochastic process is given by:*

$$F_X(x) := P\{X_t < x\} = (1 - b_c)F_\varepsilon(x) + b_c F_{\sqrt{2}\varepsilon}(x), \tag{5a}$$

*where $F_\varepsilon(x)$ and $F_{2\varepsilon}(x)$ are CDFs of RVs $\varepsilon_t : \mathcal{N}(0, \sigma^2)$ and $\sqrt{2}\varepsilon_t : \mathcal{N}(0, 2\sigma^2)$, respectively.*

**Proof.** For arbitrary $t = 0, 1, \ldots, T$, let us denote the series of RVs $\eta_t = \theta_t \varepsilon_t$. Since $\theta_t$ and $\varepsilon_t$ are mutually independent RVs, it follows

$$E(\eta_t) = E(\theta_t)E(\varepsilon_t) = 0,$$
$$V(\eta_t) = E(\theta_t^2)E(\varepsilon_t^2) = b_c\sigma^2.$$

Moreover, it is simply shown that $Cov(\eta_t, \eta_{t+h}) = 0$ holds for every $h \neq 0$, i.e., $(\eta_t)$ is a series of uncorrelated RVs. By applying conditional probabilities, the CDF of these RVs can be obtained as follows:

$$
\begin{aligned}
F_\eta(x) : &= P\{\eta_t < x\} \\
&= P\{\eta_t < x | \theta_t = 1\} \cdot P\{\theta_t = 1\} + P\{\eta_t < x | \theta_t = 0\} \cdot P\{\theta_t = 0\} \\
&= P\{\varepsilon_t < x\} \cdot P\{\theta_t = 1\} + P\{x > 0\} \cdot P\{\theta_t = 0\} \\
&= b_c F_\varepsilon(x) + (1 - b_c)F_0(x),
\end{aligned}
$$

where $F_0(x) = I(x > 0)$ is the CDF of the RV $I_0 \overset{as}{=} 0$. Based on that, for the CF of the RVs $\eta_t$, one obtains:

$$
\begin{aligned}
\varphi_\eta(u) : &= \int\limits_{-\infty}^{+\infty} e^{iux} F_\eta(dx) = \int\limits_{-\infty}^{+\infty} e^{iux}[b_c F_\varepsilon + (1 - b_c)F_0](dx) \\
&= b_c\varphi_\varepsilon(u) + (1 - b_c)\varphi_0(u).
\end{aligned}
$$

Here, $\varphi_\varepsilon(u) = e^{-\frac{\sigma^2 u^2}{2}}$ and $\varphi_0(u) \equiv 1$ are CFs of the RVs $\varepsilon_t$ и $I_0$, respectively. By substituting these CFs into the previous equality, we have:

$$\varphi_\eta(u) = 1 + b_c\left(e^{-\frac{\sigma^2 u^2}{2}} - 1\right),$$

whence, by applying Equation (4), it follows that the CF of RVs $X_t$ is:

$$
\begin{aligned}
\varphi_X(u) &= \varphi_\varepsilon(u) \cdot \varphi_\eta(u) = e^{-\frac{\sigma^2 u^2}{2}}\left[1 + b_c\left(e^{-\frac{\sigma^2 u^2}{2}} - 1\right)\right] \\
&= (1 - b_c)e^{-\frac{\sigma^2 u^2}{2}} + b_c e^{-\sigma^2 u^2}.
\end{aligned}
$$

According to the last equality and Lévy's correspondence theorem (see, e.g., [33] (p. 181)), Equation (5) immediately follows, that is, the statement of the theorem is proved. □

**Remark 1.** As shown in [12], the CDF of RVs $X_t$ can also be given in the following form:

$$F_X(x) := P\{X_t < x\} = [(1 - b_c)F_0(x) + b_c F_\varepsilon(x)] \otimes F_\varepsilon(x), \tag{5b}$$

where "$\otimes$" denotes the convolution of two (arbitrary) CDFs $F(x)$, $G(x)$:

$$(F \otimes G)(x) := \int\limits_{-\infty}^{+\infty} F(x - y)G(dy).$$

The equivalence of Equations (5a) and (5b) are directly obtained from the fact that CDF $F_0(x)$ is neutral for the convolution operator, i.e.,

$$(F \otimes F_0)(x) = (F_0 \otimes F)(x) = \int\limits_{-\infty}^{+\infty} I(x > y)F(dy) = F(x).$$

Finally, note that by differentiating Equation (5), the probability density function (PDF) of the series $(X_t)$, one obtains:

$$f_X(x) = \frac{1 - b_c}{\sigma\sqrt{2\pi}}e^{-\frac{x^2}{2\pi\sigma^2}} + \frac{b_c}{2\sigma\sqrt{\pi}}e^{-\frac{x^2}{4\pi\sigma^2}}.$$

By a similar procedure as in the previous theorem and using the convolutions of CDFs, we describe the stochastic distribution of other components of the GSB process, i.e., the series $(m_t)$ and $(y_t)$. As already shown in the previous section, these series represent non-stationary stochastic processes with a constant mean $\mu = E(m_t) = E(y_t)$. Accordingly, the following statement is valid.

**Theorem 2.** *Let $(y_t)$ and $(m_t)$ be the time series defined by Equations (1) and (2), respectively, where $m_0 \overset{as}{=} \mu$ (const). For arbitrary $x \in \mathbb{R}$ and $t = 0, \cdot 1, \dots, T$, the CDFs of these series are as follows:*

$$F_m(x, t) := P\{m_t < x\} = \overset{t}{\underset{j=1}{\otimes}} \left[(1 - b_c)F_j(x) + b_c F_0(x)\right] \otimes F_\mu(x). \tag{6}$$

$$F_y(x, t) := P\{y_t < x\} = \overset{t}{\underset{j=1}{\otimes}} \left[(1 - b_c)F_j(x) + b_c F_0(x)\right] \otimes F_\mu(x) \otimes F_\varepsilon(x). \tag{7}$$

*Here, $F_0(x)$ and $F_j(x)$ are the CDFs of previously defined RVs $I_0$ and $\varepsilon_t$, respectively, and $F_\mu(x) = F_m(x, 0)$ is the CDF of the RV $m_0 \overset{as}{=} \mu$. In addition, when $T = +\infty$, the following convergences (in distribution) are valid:*

$$\frac{1}{\sqrt{t}}m_t \overset{d}{\to} \mathcal{N}\left(0, a_c\sigma^2\right), \qquad \frac{1}{\sqrt{t}}y_t \overset{d}{\to} \mathcal{N}\left(0, a_c\sigma^2\right), \qquad t \to +\infty. \tag{8}$$

**Proof.** For arbitrary $t = 0, 1, \dots, T$, let us introduce a series of RVs $\xi_t = q_t\varepsilon_t$. In the same way as in the proof of the previous theorem, it is shown that $(\xi_t)$ is a series of mutually uncorrelated RVs, with $E(\xi_t) = 0$, $D(\xi_t) = a_c\sigma^2$, where $a_c = E(q_t) = P\{\varepsilon_t^2 > c\} = 1 - b_c$. By reapplying the conditional probabilities, the CDF of $\xi_t$ is obtained as follows:

$$
\begin{aligned}
F_\xi(x): \quad &= P\{\xi_t < x\} \\
&= P\{\xi_t < x | q_t = 1\} \cdot P\{q_t = 1\} + P\{\xi_t < x | q_t = 0\} \cdot P\{q_t = 0\} \\
&= P\{\varepsilon_t < x\} \cdot P\{q_t = 1\} + P\{x > 0\} \cdot P\{q_t = 0\} \\
&= a_c F_\varepsilon(x) + (1 - a_c)F_0(x).
\end{aligned}
$$

According to this, their corresponding CF is obtained:

$$
\begin{aligned}
\varphi_\xi(u) \quad &= \int\limits_{-\infty}^{+\infty} e^{iux} F_\xi(dx) = \int\limits_{-\infty}^{+\infty} e^{iux}[a_c F_\varepsilon + (1 - a_c)F_0](dx) \\
&= a_c\varphi_\varepsilon(u) + (1 - a_c)\varphi_0(u) = 1 + a_c\left(e^{-\frac{\sigma^2 u^2}{2}} - 1\right) \\
&= (1 - b_c)e^{-\frac{\sigma^2 u^2}{2}} + b_c.
\end{aligned}
$$

Applying Equation (2), we find that the CFs of the RVs $(m_t)$ are as follows:

$$\varphi_m(u,t) = \varphi_\mu(u) \prod_{j=0}^{t-1} \varphi_\xi(u) = e^{iu\mu} \left[ (1 - b_c) e^{-\frac{\sigma^2 u^2}{2}} + b_c \right]^t, \tag{9}$$

where $\varphi_\mu(u) = e^{iu\mu}$ is CF of the RV $m_0 \overset{as}{=} \mu$. Then, Equation (6) immediately follows from Equation (9) and Lévy's correspondence theorem [33] (p. 181).

Similarly, by applying the previous Equations (1) and (9), the CFs of the RVs $(y_t)$ are obtained:

$$\varphi_y(u,t) = \varphi_m(u) \cdot \varphi_\varepsilon(u) = e^{iu\mu - \frac{\sigma^2 u^2}{2}} \left[ (1 - b_c) e^{-\frac{\sigma^2 u^2}{2}} + b_c \right]^t. \tag{10}$$

From here, by reapplying the theorem of Lévy, Equation (7) immediately follows.

To prove the second part of the theorem, i.e., Equation (8), note first that the CFs of the RVs $m_t/\sqrt{t}$ and $y_t/\sqrt{t}$, when $t = 1, 2, \ldots$, according to Equations (9) and (10), can be written as follows:

$$
\begin{aligned}
\varphi_m\left(\frac{u}{\sqrt{t}}, t\right) &= e^{iu\mu/\sqrt{t}} \left[ 1 + a_c \left( e^{-\frac{\sigma^2 u^2}{2t}} - 1 \right) \right]^t \\
&= e^{iu\mu/\sqrt{t}} \left[ 1 - \frac{a_c \sigma^2 u^2}{2t} + \sigma\left(\frac{u^2}{t}\right) \right]^t, \\
\varphi_y\left(\frac{u}{\sqrt{t}}, t\right) &= e^{iu\mu/\sqrt{t} - \frac{\sigma^2 u^2}{2t}} \left[ 1 + a_c \left( e^{-\frac{\sigma^2 u^2}{2t}} - 1 \right) \right]^t \\
&= e^{iu\mu/\sqrt{t} - \frac{\sigma^2 u^2}{2t}} \left[ 1 - \frac{a_c \sigma^2 u^2}{2t} + \sigma\left(\frac{u^2}{t}\right) \right]^t.
\end{aligned}
$$

Here, $\sigma(z)$ is an infinitely small value of a higher order than $z$ when $z \to 0$. Hence, for a fixed but arbitrary $u \in \mathbb{R}$, we have:

$$\varphi_m\left(\frac{u}{\sqrt{t}}, t\right) \to e^{-\frac{a_c \sigma^2 u^2}{2}}, \quad \varphi_y\left(\frac{u}{\sqrt{t}}, t\right) \to e^{-\frac{a_c \sigma^2 u^2}{2}}, \quad t \to +\infty,$$

and the convergences thus obtained confirm the asymptotic relations in Equation (8). □

**Remark 2.** Note again that the proofs of the previous two theorems are based on determining the CFs of the corresponding time series of the GSB process. In this sense, the CFs of the uncorrelated series of RVs $(\xi_t)$ and $(\eta_t)$ play a fundamental role. The series $(\xi_t)$ and $(\eta_t)$ can be viewed as "new" innovations with "optional" non-zero values, which essentially describe the stochastic structure of the GSB process. Nevertheless, as the relation $\eta_t + \xi_t \overset{as}{=} \varepsilon_t$ holds for each $t = 0, \cdot 1, \ldots, T$, it is sufficient to consider only one of these two series of uncorrelated RVs (which is what was done in the statement of Theorem 2). Moreover, it can be easily shown that CDFs:

$$
\begin{aligned}
F_\xi(u) &= (1 - b_c) F_\varepsilon(x) + b_c F_0(x), \\
F_\eta(u) &= b_c F_\varepsilon(u) + (1 - b_c) F_0(u)
\end{aligned}
$$

are continuous almost everywhere, with the only point of discontinuity $x = 0$ where they have "jumps" of the values $b_c$ and $1 - b_c$, respectively (see for more detail [34,35]). Therefore, the CDFs of the series $(\xi_t)$ and $(\eta_t)$ are mixtures of Gaussian and discrete type distribution, usually named *Contaminated Gaussian Distribution (CGD)*. This is another important fact that disables an application of some of the standard procedures in the investigation of the properties of non-stationary series $(y_t)$ and $(m_t)$.

On the other hand, Equation (8) shows that even non-stationary time series $(m_t)$ and $(y_t)$ can generate series $\left(m_t/\sqrt{t}\right)$ and $\left(y_t/\sqrt{t}\right)$ that converge toward a normal distribution

when $t \to +\infty$. Moreover, based on the properties of the non-stationary components of the GSB process described in Section 2, the time series $\left( m_t / \sqrt{t} \right)$ has a constant variance $a_c \sigma^2$. These facts will be of importance in the practical application of the GSB process and can be readily observed based on the convergence of the corresponding CFs $\varphi_m \left( u / \sqrt{t}, t \right)$ and $\varphi_y \left( u / \sqrt{t}, t \right)$. As an illustration, Figure 2 shows convergences of the modulus of these CFs, for different time indices $(t)$.

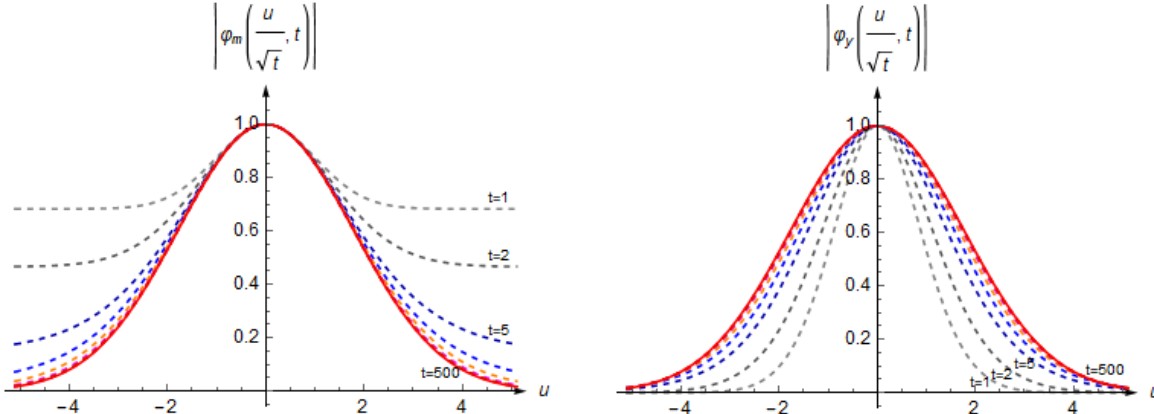

**Figure 2.** Graphs of the convergence of modulus of the characteristic functions $\varphi_m \left( u / \sqrt{t}, t \right)$ and $\varphi_y \left( u / \sqrt{t}, t \right)$, when $t = 1, 2, \ldots, 500$. (Parameter values are: $\mu = c = \sigma = 1$).

At the end of this section, we additionally describe some more asymptotic properties of series obtained by transformations of non-stationary time series $(m_t)$ and $(y_t)$. They also refer to the possibility of finding their asymptotically normal (AN) distributions, which can be shown by the following statement:

**Theorem 3.** *For arbitrary $\alpha \geq 1$ and time series $(y_t)$ and $(m_t)$, given by Equations (1) and (2), respectively, let us define the so-called $\alpha$-mean series:*

$$\overline{M}_{t;\alpha} = \frac{1}{t^\alpha} \sum_{j=1}^{t} m_j, \quad \overline{Y}_{t;\alpha} = \frac{1}{t^\alpha} \sum_{j=1}^{t} y_j,$$

*Then the following statements hold:*

*(i).　When $1 \leq \alpha \leq 3/2$, time series $\overline{M}_{t;\alpha}$ and $\overline{Y}_{t;\alpha}$ have an asymptotically normal distribution, i.e., the following relations, when $t \to +\infty$, are valid:*

$$\overline{M}_{t;\alpha} \sim \mathcal{N} \left( \mu t^{1-\alpha}, \frac{a_c \sigma^2 t^{3-2\alpha}}{3} \right), \quad \overline{Y}_{t;\alpha} \sim \mathcal{N} \left( \mu t^{1-\alpha}, \frac{a_c \sigma^2 t^{3-2\alpha}}{3} \right). \qquad (11)$$

*(ii).　When $\alpha > 3/2$, time series $\overline{M}_{t;\alpha}$ and $\overline{Y}_{t;\alpha}$ asymptotically vanish, i.e.,*

$$\overline{M}_{t;\alpha} \xrightarrow{d} I_0, \quad \overline{Y}_{t;\alpha} \xrightarrow{d} I_0, \quad t \to +\infty. \qquad (12)$$

**Proof.** We show the statement of the theorem first for the time series $\overline{M}_{t;\alpha}$. Based on the definition of time series $(m_t)$, i.e., Equation (2), one obtains:

$$
\begin{aligned}
\overline{M}_{t;\alpha} &= \frac{1}{t^\alpha} \sum_{j=1}^t m_j = \frac{1}{t^\alpha} \sum_{j=1}^t \left( m_0 + \sum_{k=0}^{j-1} q_k \varepsilon_k \right) \\
&= \frac{1}{t^\alpha} \left[ tm_0 + \sum_{j=0}^{t-1} (t-j) q_j \varepsilon_j \right] = t^{1-\alpha} m_0 + \sum_{k=1}^t \frac{k}{t^\alpha} \xi_{t-k}.
\end{aligned}
$$

Thus, the series $\overline{M}_{t;\alpha}$ is represented as a sum of uncorrelated RVs $\xi_{t-k}$, $k = 1, \ldots, t$. By applying the well-known properties of the CFs, as well as the expressions for the CF of the series $(\xi_t)$, the CFs of $\overline{M}_{t;\alpha}$ are as follows:

$$
\varphi_{\overline{M};\alpha}(u,t) = \varphi_m \left( \frac{u}{t^{\alpha-1}}, 0 \right) \prod_{k=1}^t \varphi_\xi \left( \frac{ku}{t^\alpha} \right) = e^{iu\mu t^{1-\alpha}} \prod_{k=1}^t \left[ 1 + a_c \left( e^{-\frac{k^2 \sigma^2 u^2}{2t^{2\alpha}}} - 1 \right) \right].
$$

Taking the logarithm of the function $\varphi_{\overline{M};\alpha}(u,t)$ gives a function:

$$
\psi_M(u,t,\alpha) := \ln \varphi_{\overline{M};\alpha}(u,t) = iu\mu t^{1-\alpha} + \sum_{k=1}^t f_k(u,t,\alpha),
$$

where $f_k(u,t,\alpha) := \ln\left[ 1 + a_c \left( \exp\left( -k^2 \sigma^2 u^2 t^{-2\alpha}/2 \right) - 1 \right) \right]$. After some computation, we find that, when $0 < a_c < 1$,

$$
\frac{\partial f_k(0,t,\alpha)}{\partial u} = \left. \frac{-\frac{a_c k^2 \sigma^2 u}{t^{2\alpha}} e^{-\frac{k^2 \sigma^2 u^2}{2t^{2\alpha}}}}{1 + a_c \left( e^{-\frac{k^2 \sigma^2 u^2}{2t^{2\alpha}}} - 1 \right)} \right|_{u=0} = 0
$$

$$
\frac{\partial^2 f_k(0,t,\alpha)}{\partial u^2} = \left. \frac{-\frac{a_c k^2 \sigma^2}{t^{2\alpha}} e^{-\frac{k^2 \sigma^2 u^2}{2t^{2\alpha}}} \left( (1 - a_c) \left( 1 - \frac{k^2 \sigma^2 u^2}{t^{2\alpha}} \right) + a_c e^{-\frac{k^2 \sigma^2 u^2}{2t^{2\alpha}}} \right)}{\left( 1 + a_c \left( e^{-\frac{k^2 \sigma^2 u^2}{2t^{2\alpha}}} - 1 \right) \right)^2} \right|_{u=0} = -\frac{a_c k^2 \sigma^2}{t^{2\alpha}}.
$$

Thus, the functions $f_k(u,t,\alpha)$ have local maxima at the point $u = 0$. Using a similar procedure as in [34], that is, by Laplace approximation of functions $f_k(u,t,\alpha)$ at $u = 0$, one obtains:

$$
\begin{aligned}
\psi_M(u,t,\alpha) &= iu\mu t^{1-\alpha} + \sum_{k=1}^t \left[ \frac{\partial^2 f_k(0,t,\alpha)}{\partial u^2} \cdot \frac{u^2}{2} + \sigma_k(u^2) \right] \\
&= iu\mu t^{1-\alpha} + \sum_{k=1}^t \left[ -\frac{a_c k^2 \sigma^2 u^2}{2t^{2\alpha}} + \sigma_k(t^{-2\alpha} u^2) \right] \\
&= iu\mu t^{1-\alpha} - \frac{a_c \sigma^2 u^2}{12 t^{2\alpha}} t(t+1)(2t+1) + \sigma(t^{3-2\alpha} u^2).
\end{aligned}
$$

Then, by taking the asymptotic value in the last expression, when $t \to +\infty$, it follows:

$$
\psi_M(u,t,\alpha) \sim \begin{cases} iu\mu t^{1-\alpha} - a_c \sigma^2 t^{3-2\alpha}/6, & 1 \le \alpha \le 3/2 \\ 0, & \alpha > 3/2. \end{cases}
$$

Substituting this expression into the CFs $\varphi_{\overline{M};\alpha}(u,t)$, it is easy to conclude that the first part of the theorem, in the sense of the series $\overline{M}_{t;\alpha}$, is valid.

The proof for the series $\overline{Y}_{t;\alpha}$ is carried out analogously. Using Equation (1), as the previously proven facts, we have that

$$
\begin{aligned}
\overline{Y}_{t;\alpha} &= \tfrac{1}{t^\alpha} \sum_{j=1}^{t} \left( m_j + \varepsilon_j \right) = \overline{M}_{t;\alpha} + \sum_{j=1}^{t} \tfrac{\varepsilon_j}{t^\alpha} = t^{1-\alpha} m_0 + \sum_{k=1}^{t} \tfrac{k}{t^\alpha} \xi_{t-k} + \sum_{k=0}^{t-1} \tfrac{\varepsilon_{t-k}}{t^\alpha} \\
&= t^{1-\alpha} m_0 + \tfrac{\varepsilon_t}{t^\alpha} + \sum_{k=1}^{t} \left( 1 + k q_{t-k} \right) \tfrac{\varepsilon_{t-k}}{t^\alpha}.
\end{aligned}
$$

Since RVs $\varepsilon_{t-k}$, $k = 0, 1, \ldots, t$, are mutually independent, after some computation, we obtain the CFs of series $\overline{Y}_{t;\alpha}$ as follows:

$$
\begin{aligned}
\varphi_{\overline{Y};\alpha}(u, t) &= \varphi_m \left( \tfrac{u}{t^{\alpha-1}}, 0 \right) \varphi_\varepsilon \left( \tfrac{u}{t^\alpha} \right) \prod_{k=1}^{t} \left[ (1 - a_c) \varphi_\varepsilon \left( \tfrac{u}{t^\alpha} \right) + a_c \varphi_\varepsilon \left( \tfrac{(k+1)u}{t^\alpha} \right) \right] \\
&= e^{iu\mu t^{1-\alpha} - \frac{\sigma^2 u^2}{2t^{2\alpha}}} \prod_{k=1}^{t} \left[ e^{-\frac{\sigma^2 u^2}{2t^{2\alpha}}} + a_c \left( e^{-\frac{(k+1)^2 \sigma^2 u^2}{2t^{2\alpha}}} - e^{-\frac{\sigma^2 u^2}{2t^{2\alpha}}} \right) \right] \\
&= e^{iu\mu t^{1-\alpha} - \frac{\sigma^2 u^2 (t+1)}{2t^{2\alpha}}} \prod_{k=1}^{t} \left[ 1 + a_c \left( e^{-\frac{(k^2+2k)\sigma^2 u^2}{2t^{2\alpha}}} - 1 \right) \right].
\end{aligned}
$$

From here, using the same procedure as in the previous part of the proof, i.e., by taking the logarithm of the function $\varphi_{\overline{Y};\alpha}(u, t)$, and by developing $\psi_Y(u, t, \alpha) := \ln \varphi_{\overline{Y};\alpha}(u, t)$ at the point $u = 0$, we have:

$$
\begin{aligned}
\psi_Y(u, t, \alpha) &= iu\mu t^{1-\alpha} - \tfrac{\sigma^2 u^2 (t+1)}{2t^{2\alpha}} + \sum_{k=1}^{t} \ln \left[ 1 + a_c \left( e^{-\frac{(k^2+2k)\sigma^2 u^2}{2t^{2\alpha}}} - 1 \right) \right] \\
&= iu\mu t^{1-\alpha} - \tfrac{\sigma^2 u^2 (t+1)}{2t^{2\alpha}} - \sum_{k=1}^{t} \left[ \tfrac{a_c (k^2+2k) \sigma^2 u^2}{2t^{2\alpha}} + \sigma_k \left( t^{-2\alpha} u^2 \right) \right] \\
&= iu\mu t^{1-\alpha} - \tfrac{\sigma^2 u^2}{2} \left( t^{1-2\alpha} + t^{-2\alpha} \right) - a_c \tfrac{\sigma^2 u^2}{12 t^{2\alpha}} t(t+1)(2t+7) \\
&\quad + \sigma \left( t^{3-2\alpha} u^2 \right).
\end{aligned}
$$

Finally, taking the asymptotic values, when $t \to +\infty$, one obtains:

$$
\psi_Y(u, t, \alpha) \sim
\begin{cases}
iu\mu t^{1-\alpha} - \tfrac{\sigma^2 u^2}{2} \left( t^{1-2\alpha} + t^{-2\alpha} + \tfrac{a_c t^{3-2\alpha}}{3} \right), & 1 \leq \alpha \leq 3/2 \\
0, & \alpha > 3/2.
\end{cases}
$$

Substituting this expression into CFs $\varphi_{\overline{Y};\alpha}(u, t)$, the entire statement of the theorem is proved. $\square$

**Remark 3.** In the previous theorem, the case $\alpha = 3/2$ is particularly interesting because Equation (11) then gives the following convergences:

$$
\tfrac{1}{t^{3/2}} \sum_{j=1}^{t} m_j \xrightarrow{d} \mathcal{N} \left( 0, \tfrac{a_c \sigma^2}{3} \right), \quad \tfrac{1}{t^{3/2}} \sum_{j=1}^{t} y_j \xrightarrow{d} \mathcal{N} \left( 0, \tfrac{a_c \sigma^2}{3} \right), \quad t \to +\infty. \tag{13}
$$

We will call these convergences, in the usual way, *central limit theorems (CLTs) for the GSB process*. As will be seen below, they will be helpful for estimating the unknown parameters of the GSB process, primarily the conditional variance $\sigma^2$.

## 4. Parameter Estimation Procedures

Now, let us consider the problem of estimation of (unknown) parameters of the GSB process, the critical value ($c$), mean value ($\mu$), and conditional variance ($\sigma^2$). To estimate the first parameter $c$, a series of increments ($X_t$) will be used as the (only) observable and stationary component of the GSB model. Recall that we have named this series the Split-MA (1) process because it is close to standard, linear MA models. Although some of the estimation procedures we present here are like standard estimation methods in MA models (see, for instance [36]), the specificity of the Split-MA (1) model requires additional

testing and analysis, primarily of the quality of the obtained estimates. To that end, the consistency and asymptotic normality of the estimators were examined. After that, several new approaches were considered, based on the observation of non-stationary time series $(y_t)$. The main goal of these procedures is aimed at obtaining the estimated values of the parameters $\mu$ and $\sigma^2$.

### 4.1. Estimates of Critical Value (c)

Let $(X_t)$ be the Split-MA (1) process defined by Equation (4). As we have already shown, the first correlation coefficient of this series is:

$$\rho_X(1) = -\frac{b_c}{1+b_c}, \quad 0 < b_c < 1.$$

From here, by solving on $b_c$, we get the estimated value of this parameter:

$$\widetilde{b}_c = -\frac{\hat{\rho}_X(1)}{1+\hat{\rho}(1)}, \quad 0 < b_c < 1, \tag{14}$$

where:

$$\hat{\rho}_X(1) = \left(\sum_{t=1}^{T} X_t X_{t-1}\right)\left(\sum_{t=1}^{T} X_t^2\right)^{-1}$$

is the estimated value of the first correlation. Based on the estimate $\widetilde{b}_c$, the corresponding estimate of the critical value $c = \widetilde{c}$ can be determined as a solution to the equation:

$$P\left\{\varepsilon_t^2 \le c\right\} = \widetilde{b}_c.$$

According to Equation (14), it is easy to see that $\widetilde{b}_c$ and $\widetilde{c}$ are appropriate estimates if the following inequalities hold:

$$0 < \widetilde{b}_c < 1 \quad \Longleftrightarrow \quad -0.5 < \hat{\rho}_X(1) < 0.$$

In [9], it was shown that thus obtained estimators are strictly consistent if the innovations $(\varepsilon_t)$ have a continuous distribution. Moreover, the estimates $\widetilde{b}_c$ and $\widetilde{c}$ will also be asymptotically normal (AN) if the RVs $(\varepsilon_t)$ have a symmetric distribution. Note that both conditions are fulfilled in the case of Gaussian innovations $\varepsilon_t : \mathcal{N}(0, \sigma^2)$, when the RVs $(\varepsilon_t/\sigma)^2$ have a $\chi_1^2$ distribution. Thus, the estimate of the critical value $\widetilde{c}$ is simply found from the equality:

$$\widetilde{c} = \widetilde{\sigma}^2 \cdot F_{\chi_1^2}^{-1}\left(\widetilde{b}_c\right). \tag{15}$$

Here, $\widetilde{\sigma}^2$ is the estimated variance of innovations $(\varepsilon_t)$ which will be described later.

However, it can be shown that, as for the linear MA series, the estimate $\widetilde{b}_c$ is not the most efficient estimate for $b_c$ (asymptotic efficiency of the estimate $\widetilde{b}_c$ is analyzed at the end of this subsection). To obtain more efficient estimates of the given parameters, we will modify the well-known Gauss-Newton method of estimating the parameters of nonlinear functions (see, for instance [36]). First, notice that Equation (4) can be written in the form:

$$\varepsilon_t = X_t + \theta_{t-1}\varepsilon_{t-1}, \quad t = 1, \dots, T$$

or, in functional form,

$$\varepsilon_t(X, \theta) = X_t + \theta_{t-1}\varepsilon_{t-1}(X, \theta). \tag{16}$$

On the other hand, if we define a series of RVs as

$$W_t(X, \theta) = \theta_t W_{t-1}(X, \theta) + \varepsilon_{t-1}(X, \theta), \tag{17}$$

then it is easy to see that the RVs $W_t(X, \theta)$ are $\mathcal{F}_{t-1}$ adapted, for each $t = 1, \ldots, T$, and thus independent of $\varepsilon_t$ and $\theta_{t+1}$. According to mentioned properties of RVs $(\theta_t)$ and $(\varepsilon_t)$, it follows that $(W_t(X, \theta))$ is a stationary and ergodic series of RVs (see, for more detail [37]) with $E(W_t(X, \theta)) = 0$ and correlation function $\rho_W(h) = b_c^{|h|}, h = 0, \pm 1, \ldots$ To this series, using the procedure described in [38], we add the so-called residual series:

$$R_t(X, \theta) = W_t(X, \theta) - b_c W_{t-1}(X, \theta). \tag{18}$$

The RVs $R_t(X, \theta)$ are also $\mathcal{F}_{t-1}$ adapted and mutually non-correlated, which can easily be shown. Namely, by applying Equations (16)–(18), for any integer $h > 0$, one obtains:

$$
\begin{aligned}
Cov(R_t(X, \theta), R_{t+h}(X, \theta)) &= E(R_t(X, \theta) R_{t+h}(X, \theta)) \\
&= E[R_t(X, \theta)(W_{t+h}(X, \theta) - b_c W_{t+h-1}(X, \theta))] \\
&= E(R_t(X, \theta) W_{t+h}(X, \theta)) - b_c E(R_t(X, \theta) W_{t+h-1}(X, \theta)) \\
&= E[R_t(X, \theta) \theta_{t+h} W_{t+h-1}(X, \theta)] - b_c E(R_t(X, \theta) W_{t+h-1}(X, \theta)) = 0.
\end{aligned}
$$

Thus, Equation (18) defines the series $(W_t(X, \theta))$ as a linear autoregressive (AR) process with innovations $(R_t(X, \theta))$. From here, we obtain another estimate of the unknown parameter $b_c \in (0, 1)$ by the following algorithmic procedure:

(1)  Applying Equation (14), determine $\widetilde{b}_c$ as (the initial) estimate of $b_c$, and according to Equation (15), determine estimate $\widetilde{c}$.

(2)  Based on Equations (16)–(18) and having obtained an estimate $\widetilde{b}_c$, compute, for each $t = 1, \ldots, T$, the values:

$$\widetilde{\theta}_t := I\left(\varepsilon_{t-1}^2\left(X, \widetilde{\theta}\right) \leq \widetilde{c}\right)$$

$$\varepsilon_t\left(X, \widetilde{\theta}\right) := X_t + \widetilde{\theta}_{t-1} \varepsilon_{t-1}\left(X, \widetilde{\theta}\right)$$

$$W_t\left(X, \widetilde{\theta}\right) := \widetilde{\theta}_t W_{t-1}\left(X, \widetilde{\theta}\right) + \varepsilon_{t-1}\left(X, \widetilde{\theta}\right)$$

$$R_t\left(X, \widetilde{\theta}\right) := W_t\left(X, \widetilde{\theta}\right) - \widetilde{b}_c W_{t-1}\left(X, \widetilde{\theta}\right),$$

where $\widetilde{\theta}_0 = 1, \varepsilon_0\left(X, \widetilde{\theta}\right) = \varepsilon_{-1}\left(X, \widetilde{\theta}\right) = W_0\left(X, \widetilde{\theta}\right) = 0$.

(3)  Using the standard regression procedure, i.e., the correlation function $\rho_W(h)$ when $h = 1$, obtain an estimate of $b_c$ in the form:

$$\hat{b}_c = \left(\sum_{t=0}^{T-1} W_t\left(X, \widetilde{\theta}\right) W_{t+1}\left(X, \widetilde{\theta}\right)\right) \left(\sum_{t=1}^{T} W_t^2\left(X, \widetilde{\theta}\right)\right)^{-1}.$$

(4)  As in the first step, based on the estimate $\hat{b}_c$, the critical value $\hat{c}$ can be estimated as a solution of the equation (concerning $c$):

$$P\{\varepsilon_t^2 \leq c\} = \hat{b}_c.$$

We emphasize that in [9], strict consistency and AN of the estimates $\widetilde{b}_c$ and $\widetilde{c}$ as well as $\hat{b}_c$ and $\hat{c}$ was proved. At the same time, the distribution of innovations $(\varepsilon_t)$ was not explicitly used there. In the case of GSB process, where innovations are Gaussian distributed, we can express these results as follows:

**Theorem 4.** *Estimates $\widetilde{b}_c$ and $\hat{b}_c$ are strictly consistent for the parameter $b_c$, i.e., it is valid that:*

$$\widetilde{b}_c \overset{as}{\to} b_c, \quad \hat{b}_c \overset{as}{\to} b_c, \quad T \to +\infty.$$

Moreover, the estimates $\widetilde{b}_c$ and $\hat{b}_c$ are asymptotically normal for $b_c$, i.e.,

$$\sqrt{T}\left(\widetilde{b}_c - b_c\right) \xrightarrow{d} \mathcal{N}\left(0, \widetilde{V}\right), \quad \sqrt{T}\left(\hat{b}_c - b_c\right) \xrightarrow{d} \mathcal{N}\left(0, \hat{V}\right), \quad T \to +\infty,$$

where $\widetilde{V}(b_c) = (b_c + 1)^2\left(2b_c^2 + 4b_c + 1\right)$ and $\hat{V}(b_c) = (1 - b_c)\left(3b_c^2 + 3b_c + 1\right)$.

**Remark 4.** Based on the previous theorem, the consistency and AN of the estimates $\widetilde{c}$ and $\hat{c}$, as continuous functions of $\widetilde{b}_c$ and $\hat{b}_c$, is also valid (see, for instance [9] or [39] p. 24). Additionally, for any $b_c \in (0,1)$, the inequality $\hat{V}(b_c) \leq \widetilde{V}(b_c)$ holds when the equality is valid only for $b_c = 0$, as can be seen in Figure 3. This means that asymptotic variance $\hat{V}(b_c)$, as a measure of "scattering" $\hat{b}_c$ from the true value $b_c$, is (significantly) smaller than $\widetilde{V}(b_c)$. So, $\hat{b}_c$ is a more efficient estimate than $\widetilde{b}_c$, which justifies its introduction.

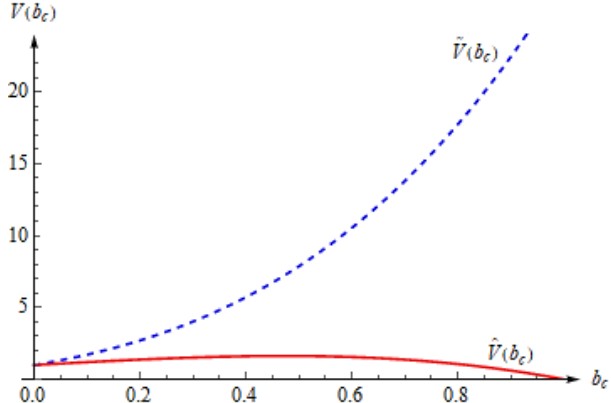

**Figure 3.** Graphs of the asymptotic variances of the estimates $\widetilde{b}_c$. (dashed line) and $\hat{b}_c$ (solid line), depending on $b_c \in (0,1)$.

*4.2. Estimates of Mean $(\mu)$*

As an estimator for the parameter $\mu = E(y_t)$, the sample mean of series $(y_t)$ was usually used:

$$\widetilde{\mu} := \overline{y}_T = \frac{1}{T}\sum_{t=1}^{T} y_t. \tag{19}$$

This estimator is obviously unbiased $E(\widetilde{\mu}) = E(\overline{y}_T) = \mu$, but its variance is not bounded. Namely, using the previously defined $\alpha$-mean series $\overline{Y}_{T;\alpha}$ when $\alpha = 1$, we can represent the estimator $\hat{\mu}$ as a sum of uncorrelated RVs:

$$\widetilde{\mu} = m_0 + \frac{1}{T}\left[\sum_{k=1}^{T}(1 + kq_{T-k})\varepsilon_{T-k} + \varepsilon_T\right].$$

Thus, for the variance of $\widetilde{\mu}$ we get:

$$
\begin{aligned}
\widetilde{V} := V(\widetilde{\mu}) \quad &= \frac{1}{T^2}\left[\sum_{k=1}^{T} V((1 + kq_{T-k})\varepsilon_{T-k}) + V(\varepsilon_T)\right] \\
&= \frac{\sigma^2}{T^2}\left[\sum_{k=1}^{T} E(1 + kq_{T-k})^2 + 1\right] \\
&= \frac{\sigma^2}{T^2}\left[\sum_{k=1}^{T}(1 + a_c k(k+2)) + 1\right] \\
&= \frac{\sigma^2}{T^2}\left[T + 1 + a_c \frac{T(T+1)(2T+7)}{6}\right] \\
&= \frac{\sigma^2(T+1)}{T^2}\left(1 + a_c \frac{T(2T+7)}{6}\right) \\
&= \frac{a_c\sigma^2 T}{3} + \mathcal{O}\left(T^{-1}\right) \to +\infty, \quad T \to +\infty.
\end{aligned}
$$

Note that, as expected, the variance $\widetilde{V} = V(\widetilde{\mu})$ is asymptotically identical to that in Theorem 3, i.e., as in Equation (11), when $\alpha = 1$. Moreover, $\widetilde{V} = 0$ when $a_c = 0$, that is, in the case of extremely large values of the parameter $c$. However, in practical applications, this condition is usually not met.

An alternative way to obtain an estimate for $\mu$ is to take the sample mean of the mean series $\overline{y}_t$, when $t = 1, \ldots, T$, i.e.,

$$\hat{\mu} := \frac{1}{T} \sum_{t=1}^{T} \overline{y}_t = \frac{1}{T} \sum_{t=1}^{T} \omega_t y_t. \tag{20}$$

Here, $\omega_t := H(T) - H(t-1)$ and $H(t) := \sum_{j=1}^{t} j^{-1}, t = 1, \ldots, T$ are the harmonic numbers, with assumption $H(0) = 0$. Obviously, $\hat{\mu}$ is also an unbiased estimate of the parameter $\mu$, but with weights that are more pronounced at the "older" points of time $(t)$ in which realizations of the series $(y_t)$ are observed. This is consistent with the fact that the covariances of RVs $y_t$ depend on these "older" time indices. Moreover, as shown in Section 2, at these time points, the covariances of RVs $y_t$ are equal to their variances. For these reasons, it is expected that the estimate $\hat{\mu}$ will be more efficient than $\widetilde{\mu}$. Indeed, using a similar procedure as before, we first represent the estimate $\hat{\mu}$ as a sum of uncorrelated RVs:

$$\begin{aligned}
\hat{\mu} &= \frac{1}{T} \sum_{t=1}^{T} \omega_t \left( m_0 + \sum_{j=0}^{t-1} q_j \varepsilon_j \right) + \frac{1}{T} \sum_{t=1}^{T} \omega_t \varepsilon_t \\
&= \frac{1}{T} \left[ m_0 \sum_{t=1}^{T} \omega_t + \sum_{j=0}^{T-1} \left( q_j \varepsilon_j \sum_{t=j+1}^{T} \omega_t \right) + \sum_{t=1}^{T} \omega_t \varepsilon_t \right].
\end{aligned}$$

As for each $j = 1, \ldots, T$, the statement below holds:

$$\sum_{t=j}^{T} \omega_t = \sum_{t=j}^{T} (H(T) - H(t-1)) = \sum_{t=j}^{T} \sum_{k=t}^{T} \frac{1}{k} = T - (j-1)(\omega_j + 1),$$

it follows that it can also be written:

$$\begin{aligned}
\hat{\mu} &= \frac{1}{T} \left[ T(m_0 + q_0 \varepsilon_0) + \sum_{j=1}^{T-1} (T - j(\omega_{j+1} + 1)) q_j \varepsilon_j \right] + \frac{1}{T} \sum_{t=1}^{T} \omega_t \varepsilon_t \\
&= m_0 + q_0 \varepsilon_0 + \frac{1}{T} \sum_{j=1}^{T-1} (c_j q_j + \omega_j) \varepsilon_j + \frac{\varepsilon_T}{T^2},
\end{aligned}$$

where $c_j = T - j(\omega_{j+1} + 1)$. Thus, after some computation, the variance of $\hat{\mu}$ one obtains is:

$$\begin{aligned}
\hat{V} := V(\hat{\mu}) &= \frac{1}{T^2} \left[ \sum_{j=1}^{T-1} E(c_j q_j + \omega_j)^2 E(\varepsilon_j^2) + \frac{E(\varepsilon_T^2)}{T^2} \right] \\
&= \frac{\sigma^2}{T^2} \left[ \sum_{j=1}^{T-1} \left( a_c c_j (c_j + 2\omega_j) + \omega_j^2 \right) + \frac{1}{T^2} \right] \\
&= \frac{\sigma^2 (a_c(T-1) - 2) H(T-1) H(T)}{T} + \sigma \left( H^{-2}(T) \right) \\
&= a_c \sigma^2 H^2(T) + \sigma \left( H^{-2}(T) \right) \to +\infty, \ T \to +\infty.
\end{aligned}$$

Notice that the variance of $\hat{V} := V(\hat{\mu})$ is also unbounded, but with a lower asymptotic order than $\widetilde{V} = V(\widetilde{\mu})$, since:

$$\lim_{T \to +\infty} \frac{V(\hat{\mu})}{V(\widetilde{\mu})} = \lim_{T \to +\infty} \frac{H^2(T)}{T} = 0.$$

This means that the estimate $\hat{\mu}$ is (asymptotically) more efficient than $\widetilde{\mu}$, which can be seen in Figure 4. Here are shown 3D plots of both variances $\widetilde{V}$ and $\hat{V}$, which were observed as functions of two variables $a_c \in (0,1)$ and $T > 0$.

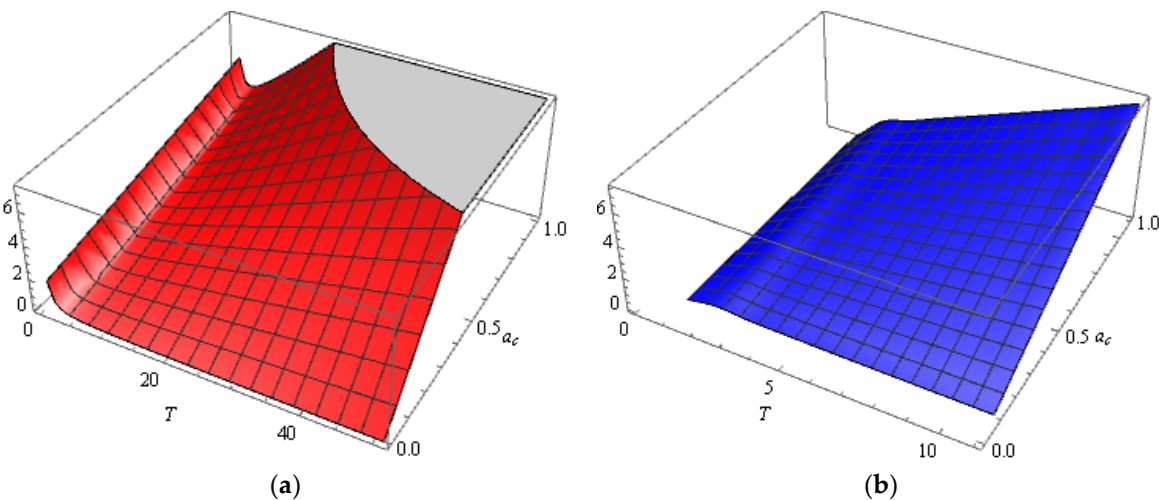

(a)　　　　　　　　　　　　　　　　　　(b)

**Figure 4.** Variances shown as 3D plots of the estimate $\widetilde{\mu}$ (**a**) and estimate $\hat{\mu}$ (**b**), depending on $a_c \in (0,1)$ and $T > 0$. (The variance of innovations is $\sigma^2 = 1$).

### 4.3. Estimates of Variance $(\sigma^2)$

Let us consider determining the estimates of the third unknown parameter $\sigma^2$, which represents the variance of the innovations $(\varepsilon_t)$, that is, the conditional variance of the base series $(y_t)$. It is precisely these facts that enable different estimation procedures for the parameter $\sigma^2$. First, notice that based on the previously obtained estimates $\widetilde{b}_c$ and $\hat{b}_c$, i.e., the modeled innovation values $(\varepsilon_t)$ given by Equation (16), the variance $\sigma^2$ can be easily estimated. The usual estimation procedure is based on sampling variance:

$$\widetilde{\sigma}^2 = \frac{1}{T} \sum_{t=1}^{T} \varepsilon_t^2\left(X, \widetilde{\theta}\right) \text{ or } \hat{\sigma}^2 = \frac{1}{T} \sum_{t=1}^{T} \varepsilon_t^2(X, \hat{\theta}). \tag{21}$$

Here, $\varepsilon_t\left(X, \widetilde{\theta}\right)$ are $\varepsilon_t(X, \hat{\theta})$ modeled innovation values obtained from the estimates $\widetilde{b}_c$ and $\hat{b}_c$, respectively. Notice that in the case of Gaussian innovations $(\varepsilon_t)$, the estimates given by Equation (21) are identical to the maximum likelihood estimators. Indeed, the log-likelihood function then reads as follows:

$$L(y_1, \ldots, y_T; \sigma^2) = -\frac{T}{2} \ln(2\pi\sigma^2) - \frac{1}{2\sigma^2} \sum_{t=1}^{T} (y_t - m_t)^2 \,,$$

and by solving the equation $\partial L(y_1, \ldots, y_T; \sigma^2)/\partial \sigma^2 = 0$, the estimate of $\sigma^2$ is obtained as in Equation (21), that is, as the sample variance of the series $(\varepsilon_t)$. Thus, the consistency and AN of both estimates $\widetilde{\sigma}^2$ and $\hat{\sigma}^2$ can be readily shown. We note that due to their equivalence, only the estimate $\hat{\sigma}^2$ will be further considered (see Theorem below).

On the other hand, note that the previous estimation procedure is based on unobservable, modeled values of innovations $(\varepsilon_t)$. Another approach to estimating the variance $\sigma^2$ is based on the so-called two-stage procedure, using the previously estimated parameter $\hat{b}_c$. By applying the equality $V(X_t) = E(X_t^2) = \sigma^2(b_c + 1)$, as well as the sample variance of the series $(X_t)$, we can obtain an estimate:

$$\hat{\sigma}_X^2 = \frac{1}{T\left(\hat{b}_c + 1\right)} \sum_{t=1}^{T} X_t^2. \tag{22}$$

Then, it follows:

**Theorem 5.** *Estimates $\hat{\sigma}^2$ and $\hat{\sigma}_X^2$ are strictly consistent for the parameter $\sigma^2$, i.e., it is valid to put:*

$$\hat{\sigma}^2 \overset{as}{\to} \sigma^2, \quad \hat{\sigma}_X^2 \overset{as}{\to} \sigma^2, \quad T \to +\infty.$$

*Moreover, the estimates $\hat{\sigma}^2$ and $\hat{\sigma}_X^2$ are asymptotically normal for $\sigma^2$, i.e.,*

$$\sqrt{T}\left(\hat{\sigma}^2 - \sigma^2\right) \overset{d}{\to} \mathcal{N}(0, V_1), \quad \sqrt{T}\left(\hat{\sigma}_X^2 - \sigma^2\right) \overset{d}{\to} \mathcal{N}(0, V_2), \quad T \to +\infty, \tag{23}$$

*where $V_1 = 2\sigma^4$ and $V_2 = \sigma^4\left(2 + 11 b_c - b_c^2\right)\left(1 + 2b_c - 3b_c^3\right)^{-1}$.*

**Proof.** Since $\left(\varepsilon_t^2\right)$ is an IID series of RVs, the stationarity and ergodicity of this series are apparent. Applying the strong low of large numbers (SLLS), it follows:

$$\hat{\sigma}^2 = \frac{1}{T}\sum_{t=1}^{T}\varepsilon_t^2\left(X, \hat{\theta}\right) \overset{as}{\to} \sigma^2.$$

Furthermore, it can easily be shown that $V\left(\hat{\sigma}^2\right) = 2\sigma^4/T$ is the variance of the estimate $\hat{\sigma}^2$. Thus, applying the central limit theorem (CLT), the first convergence in Equation (23) is obtained.

To prove the properties of the estimate $\hat{\sigma}_X^2$, we note that $\left(X_t^2\right)$ is also a stationary and ergodic series of RVs. If SLLS is now applied to the following statistics:

$$\overline{X}_t^2 := \frac{1}{T}\sum_{t=1}^{T}X_t^2, \tag{24}$$

then one obtains:

$$\frac{1}{T}\sum_{t=1}^{T}X_t^2 \overset{as}{\to} \sigma^2(b_c + 1).$$

At the same time, according to Theorem 4, we have that $\hat{b}_c$ is a strongly consistent estimator of $b_c$, i.e., $\hat{b}_c + 1 \overset{as}{\to} b_c + 1$, when $T \to +\infty$. Thus, the last two convergences give:

$$\hat{\sigma}_X^2 = \frac{\overline{X}_t^2}{\hat{b}_c + 1} \overset{as}{\to} \sigma^2, \quad T \to +\infty.$$

To prove the AN of the estimate $\hat{\sigma}_X^2$, note first that the sequence $\left(X_t^2\right)$ is 1-dependent, in the sense of Definition 6.3.1 in [36] (p. 245). According to Cauchy-Swarz and Minkowski inequalities, applied to Equation (4), i.e., the sixth moment of the sum $X_t = \varepsilon_t + (-\theta_{t-1}\varepsilon_{t-1})$, it follows that:

$$E|X_t|^6 \leq \left[\left(E|\varepsilon_t|^6\right)^{1/6} + \left(b_c\, E|\varepsilon_{t-1}|^6\right)^{1/6}\right]^6$$
$$\leq 15\sigma^6\left(1 + b_c^{1/6}\right)^6 < +\infty.$$

Then, the Hoeffding-Robbins theorem [40] can be applied, based on which it follows:

$$\sqrt{T}\overline{X}_t^2 = T^{-1/2}\sum_{t=1}^{T}X_t^2 \overset{d}{\to} \mathcal{N}\left(\sigma^2(b_c + 1), V_0\right), \tag{25}$$

for which:

$$
\begin{aligned}
V_0 &= V(X_t^2) + 2Cov(X_t^2, X_{t+1}^2) = E(X_t^4) + 2E(X_t^2 X_{t+1}^2) - 3\sigma^4(1+b_c)^2 \\
&= 3\sigma^4(1+3b_c) + 2\sigma^4(1+4b_c+b_c^2) - 3\sigma^4(1+b_c)^2 \\
&= \sigma^4(2+11b_c-b_c^2).
\end{aligned}
$$

By applying the almost sure convergence of the estimate $\hat{b}_c$ and the previously obtained convergence in Equation (25), we have

$$
\sqrt{T}\hat{\sigma}_X^2 = \frac{\sqrt{T}\overline{X_t^2}}{\hat{b}_c+1} \xrightarrow{d} \mathcal{N}(\sigma^2, V_2), \quad T \to +\infty,
$$

where $V_2 = V_0/\hat{V}(b_c)$. Thus, according to Theorem 4, the second convergence in Equation (23) is obtained. □

**Remark 5.** As in Theorem 4, by comparing the asymptotic variances $V_1$ and $V_2$ for the estimates $\hat{\sigma}^2$ and $\hat{\sigma}_X^2$, respectively, it is easy to see that inequality $V_1 \le V_2$ holds. At the same time, the equality $V_1 = V_2 = 2\sigma^4$ is valid only when $b_c = 0$ (Figure 5a), so the estimator $\hat{\sigma}^2$ is more efficient than $\hat{\sigma}_X^2$.

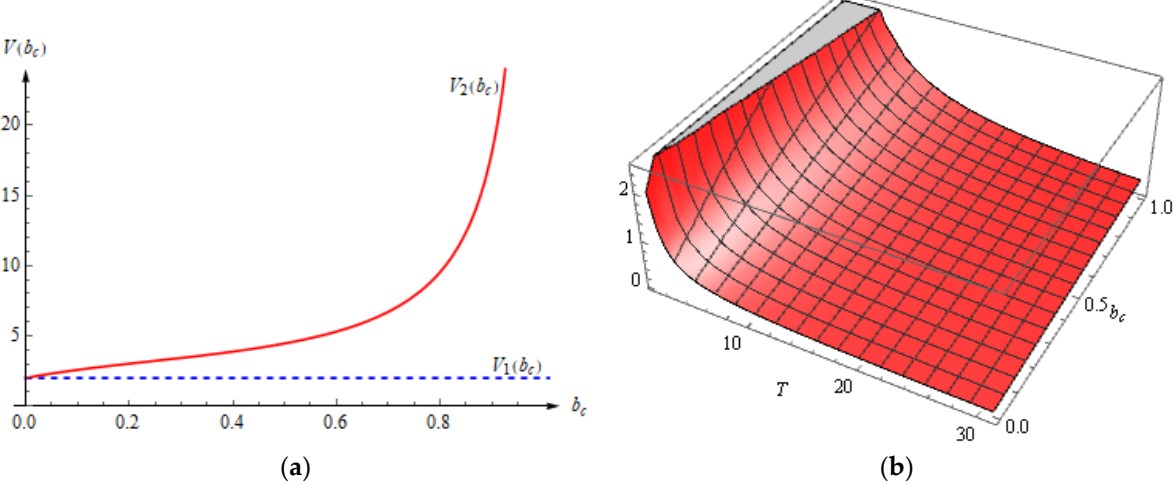

**Figure 5.** (a) Graphs of the asymptotic variances of the estimates $\hat{\sigma}^2$ (dashed line) and $\hat{\sigma}_X^2$ (solid line), depending on $b_c \in (0,1)$. (b) Plot in 3D of the variance of statistics $\overline{X_t^2}$, depending on $b_c \in (0,1)$ and $T > 0$. (The variance of the innovations is $\sigma^2 = 1$).

However, according to the proof of the previous theorem, it can be easily seen that for the variance of the statistics $\overline{X_t^2}$, given by Equation (24), is valid (Figure 5b):

$$
V(\overline{X_t^2}) = \frac{\sigma^4(2+11b_c-b_c^2)}{T} \to 0, \quad T \to +\infty.
$$

Thus, $\overline{X_t^2}$ can be used as an estimator of the "hybrid" parameter $\sigma^2(b_c+1)$, which will be of interest for practical research, that is, the application of the GSB model discussed below.

Finally, another approach to finding estimates of the variance $\sigma^2$ is based on the observations of the non-stationary series $(y_t)$. Applying Theorem 3, i.e., the previously proven convergence in Equation (13), we have:

$$
\overline{Y}_{T;3/2} := \frac{1}{T^{3/2}} \sum_{t=1}^{T} y_t \xrightarrow{d} \mathcal{N}\left(0, \frac{a_c\sigma^2}{3}\right), \quad T \to +\infty.
$$

If we now consider the statistics:

$$S_T^2 := \overline{Y}_{T;3/2}^2 = \frac{1}{T^3}\left(\sum_{t=1}^{T} y_t\right)^2 = \frac{1}{T^3}\sum_{j=1}^{T}\sum_{k=1}^{T} y_j y_k, \tag{26}$$

after some computation, one obtains:

$$
\begin{aligned}
E(S_T^2) &= \frac{1}{T^3}\sum_{j=1}^{T}\sum_{k=1}^{T} E(y_j y_k) = \frac{1}{T^3}\sum_{j=1}^{T}\sum_{k=1}^{T}\left[Cov(y_j y_k) + \mu^2\right] \\
&= \frac{1}{T^3}\sum_{j=1}^{T}\sum_{k=1}^{T}\left[\sigma^2(\min\{j,k\}a_c + 1) + \mu^2\right] \\
&= \frac{\sigma^2}{T^3}\left[a_c\sum_{j=1}^{T}\left(j + 2\sum_{k=1}^{j-1} k\right) + T^2\right] + \frac{\mu^2}{T} = \frac{\sigma^2}{T^3}\left(a_c\sum_{j=1}^{T} j^2 + T^2\right) + \frac{\mu^2}{T} \\
&= \frac{\sigma^2 a_c}{6T^2}(T+1)(2T+1) + \frac{\sigma^2+\mu^2}{T} \to \frac{a_c \sigma^2}{3}, \quad T \to +\infty.
\end{aligned}
$$

Thus, $S_T^2$ is an asymptotically unbiased estimator for $a_c\sigma^2/3$, and using the estimate $\hat{a}_c = 1 - \hat{b}_c$, an estimator of the parameter $\sigma^2$ can be taken as:

$$\hat{\sigma}_Y^2 := \frac{3}{\hat{a}_c} S_T^2 = \frac{3}{\hat{a}_c T^3}\sum_{j=1}^{T}\sum_{k=1}^{T} y_j y_k. \tag{27}$$

## 5. Numerical Simulation and Application of the GSB Process

As already mentioned in the introductory section, two important aspects related to the practical implementation of the GSB process will be explored here. Firstly, numerical Monte Carlo simulations of previously obtained GSB estimators are analyzed. Then, based on actual data, the GSB process was applied to analyze the dynamics and distribution of the infected and immunized population with respect to COVID-19 disease in the territory of the Republic of Serbia.

### 5.1. Numerical Simulations of GSB Estimators

We first describe a pseudo-algorithm for estimating the parameters of the GSB model based on $N = 1000$ independent Monte Carlo replications of the GSB series. To that end, we assume that all series have size $T = 500$, which is close to the length of the actual series to be considered below. The primary aim is to examine the convergence, i.e., the quality of the previously proposed estimators on a sample of a given length. Therefore, corresponding estimation errors will also be investigated for this purpose. Using the previously presented theoretical facts, the pseudo-algorithm for estimating the parameters of the GSB process can be formulated as follows:

1. In the first estimation step, compute the sample correlation $\hat{\rho}_X(1)$ for a series of increments $(X_t)$. If the condition $-0.5 < \hat{\rho}_X(1) < 0$ is fulfilled, by using Equation (14), the estimator $\tilde{b}_c$ can be obtained.
2. Compute statistics $\overline{X}_t^2$, given by Equation (24), as an estimate of the "hybrid" parameter $\sigma^2(b_c + 1)$. The following variance estimator is then obtained:

$$\hat{\sigma}_X^2 = \frac{\overline{X}_t^2}{\tilde{b}_c + 1}.$$

3. According to Equation (15) and previously obtained estimates $\tilde{b}_c$ and $\hat{\sigma}_X^2$, compute the estimator $\tilde{c} = \hat{\sigma}_X^2 \cdot F_{\chi_1^2}^{-1}\left(\tilde{b}_c\right)$.

4. By using the estimate $\widetilde{c}$, for each $t = 1, \ldots, T$, generate the (modeled) values of series $(\varepsilon_t)$ and $(m_t)$, by applying the iterative procedure:

$$\begin{cases} \varepsilon_t = y_t - m_t, \\ m_t = m_{t-1} + \varepsilon_{t-1} I\{\varepsilon_{t-2}^2 \geq \widetilde{c}\}, \end{cases} \qquad (28)$$

where $\varepsilon_0 = \varepsilon_{-1} = 0$, and $m_0 = y_0 = \hat{\mu}$ is given by Equation (20).

5. According to previously obtained series $(\varepsilon_t)$, and by using Equation (21), compute a (more efficient) variance estimator $\widetilde{\sigma}^2$.

6. By applying the Gauss-Newton procedure, i.e., Equations (16)–(18), the estimate $\hat{b}_c$ can be obtained.

7. According to previously obtained estimates $\hat{b}_c$ and $\widetilde{\sigma}^2$, compute the estimator $\hat{c} = \widetilde{\sigma}^2 \cdot F_{\chi_1^2}^{-1}\left(\hat{b}_c\right)$.

We point out that in the above-mentioned pseudo algorithm, the 2nd stage can be replaced by the following alternative step:

2′.Compute statistics $S_T^2$, given by Equation (26), and estimate the "hybrid" parameter $a_c \sigma^2 / 3$. Then, according to Equation (27), the variance $\sigma^2$ can be estimated as:

$$\hat{\sigma}_Y^2 := \frac{3}{\widetilde{a}_c} S_T^2,$$

where $\widetilde{a}_c = 1 - \widetilde{b}_c$.

By applying this pseudo-algorithm, the obtained values of the estimated parameters can be summarized as shown in Table 1, where their average values (Mean), minimums (Min.), maximums (Max.) can also be seen, along with the appropriate mean squared errors of estimation (MSEE) given in parentheses. Furthermore, testing results concerning the AN of thus obtained estimates are also presented in Table 1. To that end, Anderson-Darling and Cramer-von Mises normality tests were used. Their test statistics (denoted as AD and W, respectively), as well as their corresponding $p$-values, were calculated using procedures from the R-package "nortest" [41].

According to the obtained values, it is evident that most estimators have a property of the AN. This applies even to the estimates of the mean value $\widetilde{\mu}$ and $\hat{\mu}$, which are obtained from realizations of non-stationary GSB-series $(y_t)$. As already explained, this is related to Theorems 2 and 3, which respectively describe the AN properties of the series $\left(y_t / \sqrt{t}\right)$ and so-called $\alpha$-means series. Notice that the asymptotic variance of these estimators is not bounded, hence there is a large range of their observed values. On the other hand, the AN property is not particularly emphasized in the case where the critical value $(c)$ is estimated. This is because both estimates $\widetilde{c}$ and $\hat{c}$ are obtained by the three-step procedure: estimates for the parameters $b_c$ and $\sigma^2$ should first be determined, and only then for $c$. In the case of variance estimators $\widetilde{\sigma}^2$ and $\hat{\sigma}^2$, obtained based on modeled innovations $(\varepsilon_t)$, it is easy to see that they have the highest and almost the same efficiency. Furthermore, the values of the estimator $\hat{\sigma}_X^2$ are only slightly "weaker" than $\widetilde{\sigma}^2$ and $\hat{\sigma}^2$. This is expected since, according to Theorem 5, the AN property holds for all these variance estimators. However, the estimate $\hat{\sigma}_Y^2$ is by far the weakest variance estimate and can be omitted from further analysis. Moreover, based on previously obtained theoretical results, also confirmed through simulations, the most robust estimates of the unknown parameters $c$, $\mu$,$\sigma^2$ are $\hat{c}$, $\hat{\mu}$,$\hat{\sigma}^2$, respectively. For those reasons, these estimators will be used for GSB modeling of actual data on COVID-19, which will be discussed below.

**Table 1.** Summary statistics of estimated parameters of the GSB process, obtained by a Monte Carlo study, along with realized statistics of normality tests.

| Parameters Estimators | Statistics | Values | AD (*p*-Value) | W (*p*-Value) |
|---|---|---|---|---|
| Mean ($\widetilde{\mu}$) | Min. | −24.9395 | 0.2886 (0.6161) | 0.0415 (0.6545) |
| | Mean | −0.0192 | | |
| | (MSEE) | (7.2791) | | |
| | Max. | 26.8691 | | |
| Mean ($\hat{\mu}$) | Min. | −20.0310 | 0.3363 (0.5056) | 0.0453 (0.5845) |
| | Mean | −0.00806 | | |
| | (MSEE) | (4.6055) | | |
| | Max. | 19.7987 | | |
| Critical value ($\widetilde{c}$) | Min. | 0.3849 | 1.0160 * (0.0112) | 0.1449 * (0.0278) |
| | Mean | 1.0904 | | |
| | (MSEE) | (0.5069) | | |
| | Max. | 1.6481 | | |
| Critical value ($\hat{c}$) | Min. | 0.5105 | 0.5647 (0.1435) | 0.1074 (0.0889) |
| | Mean | 0.9844 | | |
| | (MSEE) | (0.1587) | | |
| | Max. | 1.5033 | | |
| Variance ($\widetilde{\sigma}_2$) | Min. | 0.8271 | 0.3144 (0.5446) | 0.0494 (0.5182) |
| | Mean | 0.9991 | | |
| | (MSEE) | (0.0630) | | |
| | Max. | 1.2182 | | |
| Variance ($\hat{\sigma}_2$) | Min. | 0.8248 | 0.3247 (0.5231) | 0.0546 (0.4459) |
| | Mean | 1.0002 | | |
| | (MSEE) | (0.0631) | | |
| | Max. | 1.2118 | | |
| Variance ($\hat{\sigma}_Y^2$) | Min. | 0.7796 | 0.4018 (0.3584) | 0.0588 (0.3921) |
| | Mean | 1.0034 | | |
| | (MSEE) | (0.0842) | | |
| | Max. | 1.3340 | | |
| Variance ($\hat{\sigma}_X^2$) | Min. | 0.1104 | 90.626 ** (<2.2 × 10$^{-16}$) | 16.522 ** (7.37 × 10$^{-10}$) |
| | Mean | 1.0937 | | |
| | (MSEE) | (1.4183) | | |
| | Max. | 1.6313 | | |

\* $p < 0.05$, \*\* $p < 0.01$.

### 5.2. Application of the GSB Process: A Case Study of COVID-19 Dynamics

In this section we give, as an illustration, a practical application of the GSB process in stochastic modeling of actual data. In other words, as mentioned in the introductory section, we will show that it can be an adequate stochastic model for describing the dynamics of the infected and vaccinated population in relation to the SARS-CoV2 virus on the territory of the Republic of Serbia. To that end, we observe realizations of two time series ($U_t$) and ($V_t$) which, daily, represents the total number of infected persons, i.e., persons vaccinated with the first dose of the vaccine, starting from 24 December 2020 (the start date of vaccination in Serbia) and ending with 6 June 2022. The dynamics of both time series, length $T = 529$, are shown in Figure 6.

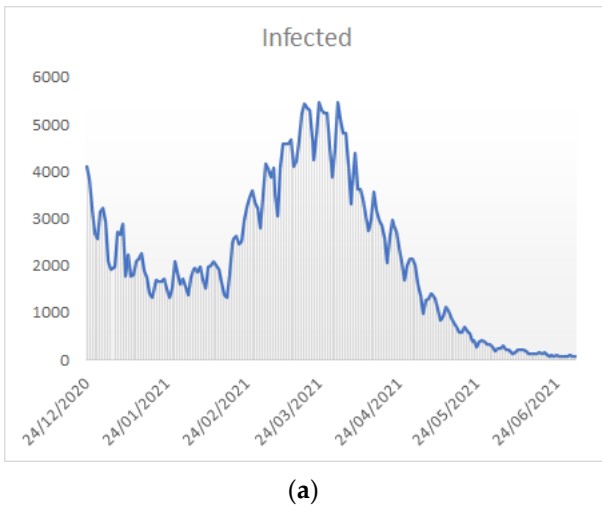
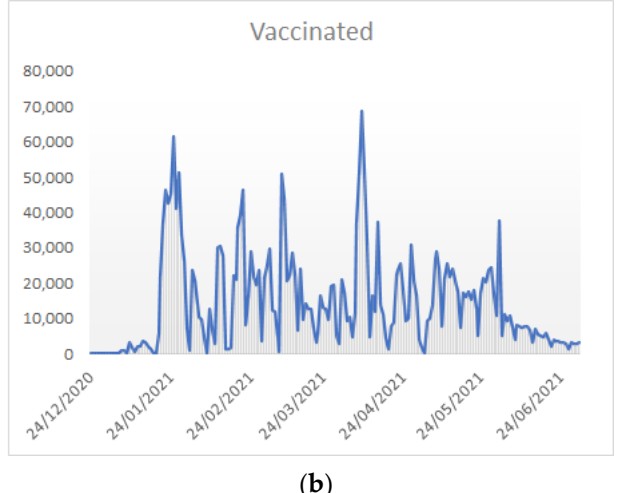

(**a**)                                             (**b**)

**Figure 6.** Dynamics of the total infected (**a**) and vaccinated population (**b**) in relation to the virus SARS-CoV2 on the territory of the Republic of Serbia.

The main statistical indicators of these series (also labeled as Series A and Series B, respectively) are shown in the following Table 2. Based on thus obtained values, it can be concluded that these are time series with distinct, pronounced fluctuations. For instance, the average number of infected people is (approximately) 3650 per day, ranging from 60 to 19,901 infected people. Similar to that, the average number of vaccinated persons is 6348 per day, but the range of vaccinated persons varies from only 4 to as many as 68,678 persons per day. Therefore, we further consider the possibility that the GSB process can be used here as an appropriate stochastic model. For this purpose, as basic sequences, we observe the realizations of the so-called *log-volumes*, i.e., logarithmic values of series $(U_t)$ and $(V_t)$:

$$y_t^{(1)} := \ln(U_t), \;\; y_t^{(2)} := \ln(V_t), \;\; t = 0, 1, \ldots, T. \tag{29}$$

Notice that the main goal of this transformation is to obtain more evenly distributed values of both series, and although based on increasing of the logarithmic function, the emphasis of fluctuations will remain. Additionally, inequalities $U_t, V_t \geq 1$ implies the non-negativity of both log-volumes series $\left(y_t^{(1)}, y_t^{(2)} \geq 0\right)$.

**Table 2.** Basic statistical indicators of observed actual series.

| Statistics | Infected (A) | Vaccinated (B) |
|:---:|:---:|:---:|
| Mean | 3650.84 | 6336 |
| Median | 2000 | 2960 |
| Mode | 1366 | 45 |
| Stand. deviation | 3650.84 | 1026.38 |
| Minimum | 60 | 4 |
| Maximum | 19,901 | 68,678 |
| Kurtosis | 8.1189 | 8.2609 |
| Skewness | 2.1418 | 2.7009 |

Further, using the log-volumes as a basic series, and using Equation (3), the series of increments $\left(X_t^{(1)}\right), \left(X_t^{(2)}\right)$ are determined entirely. Based on them, the estimates of GSB process parameters can be obtained by applying the pseudo-algorithm presented above. We emphasize that here the estimation procedure is repeated twice, i.e., for both series (A

and B). Thus, modeled values of martingale means and innovations series, generated by Equation (29), are as follows:

$$
\begin{cases}
\varepsilon_t^{(j)} = y_t^{(j)} - m_t^{(j)}, \\
m_t^{(j)} = m_{t-1}^{(j)} + \varepsilon_{t-1}^{(j)} I\left\{ \left(\varepsilon_{t-2}^{(j)}\right)^2 \geq \tilde{c} \right\},
\end{cases}
\tag{30}
$$

where $j = 1, 2$. As initial values of the iterative procedure (30), as before, we have taken $\varepsilon_0^{(j)} = \varepsilon_{-1}^{(j)} = 0$, as well as $m_0^{(j)} = y_0^{(j)} = \hat{\mu}$. Table 3 contains the basic statistical indicators of the actual series, log-volumes ($y_t^{(j)}$) and increments $\left(X_t^{(j)}\right)$, as well as modeled series, martingale means $\left(m_t^{(j)}\right)$ and innovations $\left(\varepsilon_t^{(j)}\right)$.

**Table 3.** Basic statistical indicators of actual and modeled series.

| Statistics | Series A | | | | Series B | | | |
|---|---|---|---|---|---|---|---|---|
| | $y_t^{(1)}$ | $X_t^{(1)}$ | $m_t^{(1)}$ | $\varepsilon_t^{(1)}$ | $y_t^{(2)}$ | $X_t^{(2)}$ | $m_t^{(2)}$ | $\varepsilon_t^{(2)}$ |
| Mean | 7.4041 | −0.0033 | 7.4111 | −0.0054 | 7.3544 | −0.0068 | 8.9349 | −0.1769 |
| Median | 7.5976 | −0.0336 | 7.6061 | −0.0332 | 7.9940 | −0.0566 | 9.4269 | −0.1106 |
| Stand. deviation | 1.3247 | 0.1948 | 1.3244 | 0.1912 | 2.0546 | 1.0036 | 1.7589 | 1.0238 |
| Minimum | 4.0943 | −0.5990 | 4.0943 | −0.5990 | 1.3863 | −5.0554 | 1.0986 | −6.6837 |
| Maximum | 9.8985 | 0.9125 | 9.8985 | 0.7390 | 11.1372 | 5.5147 | 11.3099 | 4.5209 |
| Kurtosis | 2.3419 | 4.3332 | 2.3305 | 3.7214 | 2.4071 | 10.1761 | 3.6732 | 10.2208 |
| Skewness | −0.5493 | 0.6114 | −0.5605 | 0.4518 | −0.4958 | 0.4290 | −1.0703 | −0.1625 |

By analyzing thus obtained values, an interesting connection can be observed, which can be explained by the previous theoretical results. Firstly, the average values of the log-volumes are "close" to the averages of the martingale means, which is in accordance with the equality $E(y_t) = E(m_t)$. Moreover, with series A, almost equal values of other statistical indicators (standard deviations, for instance) are noticeable. This can also be seen by comparing the corresponding statistical indicators of increments $\left(X_t^{(1)}\right)$ and innovations $\left(\varepsilon_t^{(1)}\right)$, which will be explained below. Table 4 shows the above-mentioned estimators obtained according to the previously described procedures. In addition, some other estimates are shown, such as the sample linear correlation $\hat{\rho}_X(1)$ and estimates of the value $b_c$. Accordingly, note that the condition $-0.5 < \hat{\rho}_X(1) < 0$ is fulfilled in the cases of both series. Moreover, let us notice, for instance, that the estimated values for $\sigma^2$ in the case of Series B are "close" to unity, so it can be assumed that innovations $(\varepsilon_t)$ in this case have a standard $\mathcal{N}(0,1)$ distribution.

**Table 4.** Estimated values of GSB process parameters.

| Parameters | Series A | Series B |
|---|---|---|
| $\tilde{\mu}$ | 7.4041 | 7.3544 |
| $\hat{\mu}$ | 7.4454 | 8.1409 |
| $\hat{\rho}_X(1)$ | −0.0126 | −0.2577 |
| $\tilde{b}_c$ | 0.0127 | 0.3472 |
| $\tilde{c}$ | 0.0003 | 0.2118 |
| $\hat{b}_c$ | 0.0953 | 0.4436 |
| $\hat{c}$ | 0.0006 | 0.3477 |
| $\tilde{\sigma}^2$ | 0.0413 | 1.0462 |
| $\hat{\sigma}^2$ | 0.0403 | 1.0634 |
| $\hat{\sigma}_X^2$ | 0.0375 | 1.0053 |

As we have already pointed out, the most robust estimators of the GSB process are $\hat{c}, \hat{\mu}, \hat{\sigma}^2$ and based on them, modeled values of the series $(m_t^{(j)})$ and $(\varepsilon_t^{(j)})$ were obtained. Let

us recall that these series, respectively, represent the stability and the impact of fluctuations in the dynamics of the total number of infected and vaccinated people. The agreement between the modeled series and the actual data can be seen in Figure 7a where, along with the empirical values of the log-volumes $(y_t^{(j)})$, modeled values of martingale means $(m_t^{(j)})$ are given. On the other hand, the agreement of a series of increments, i.e., the Split-MA(1) process $(X_t^{(j)})$ with innovations $(\varepsilon_t^{(j)})$ is shown in Figure 7b.

It should also be noted that the high agreement between the actual and modeled series is particularly noticeable in the case of series A. This can be explained theoretically, in the way it was done in Section 2. If at some points in time, innovations $(\varepsilon_t^{(1)})$ have a pronounced fluctuation, they become equal to increments $(X_t^{(1)})$ at the next moment. The agreement between the realizations of these two series will be all the better if, in addition to large and pronounced fluctuations of $(\varepsilon_t^{(1)})$, the critical value $c$ is relatively small. Note that this is precisely the case with series A, where "small" estimated values of the parameter $c$ indicate the possibility that the true value of this parameter is $c = 0$ (or, equivalently, $b_c = 0$). If the sample size is large enough, this assumption can be formally tested by the null hypothesis $H_0 : c = 0$ or, equivalently, $H_0 : b_c = 0$. According to Theorem 4, testing procedures can be based on the normal distribution, that is, using some standard, well-known statistical tests.

Note that in that case, the series of increments $(X_t^{(1)})$ is equalized with innovations $(\varepsilon_t^{(1)})$. This implies that $(y_t^{(j)})$ is a series with independent increments, i.e.,

$$X_t^{(1)} = y_t^{(1)} - y_{t-1}^{(1)} = \varepsilon_t^{(1)} \iff y_t^{(1)} = y_{t-1}^{(1)} + \varepsilon_t^{(1)}. \tag{31}$$

According to Equation (1), it follows that $y_{t-1}^{(1)} = m_t^{(1)}$, so all "information from the past" is contained in the previous realization of the series $(y_t^{(1)})$. In that way, the entire statistical analysis of this series, i.e., the dynamics of the infected population, gains simplicity; namely, series A then has (only) two stochastic components $(y_t^{(1)})$ and $(\varepsilon_t^{(1)})$, i.e., it represents a random walk series.

Finally, using the inverse transformations of those given in Equation (29), PDFs of actual series $(U_t)$ and $(V_t)$ are readily obtained:

$$f_U(x,t) = \frac{1}{x} f_y^{(1)}(\ln x, t), \quad f_V(x,t) = \frac{1}{x} f_y^{(2)}(\ln x, t). \tag{32}$$

Here, $f_y^{(j)}(\ln x, t), j = 1, 2$ are the PDFs of log-volumes $(y_t^{(j)})$, obtained by differentiating the CDFs given by Equation (9), which can be done simply. Still, due to the non-stationarity of the mentioned series, which also depends on time, it is necessary to apply some numerical procedures to calculate their PDFs. For this purpose, the R-package "distr" [42] has been used, and the results of the applied procedure are shown in Figure 8.

Here are the empirical distributions, i.e., histograms of the number of infected and vaccinated persons per day, with their fitted PDFs, obtained using Equations (32). Due to the non-stationarity of the time series $(U_t)$ and $(V_t)$, as well as the comparison of the theoretical PDFs, fitting was also performed for the PDFs $f_U(x,t)$ and $f_V(x,t)$ of length $t = 50, 10, \ldots, 500 < T = 529$ (shown with dashed lines in Figure 8). In the case of the infected population (Series A), according to Equation (31) and the condition $c \approx 0$, it follows that RVs $y_t^{(1)}$ have (an approximately) normal $\mathcal{N}(\mu, (t+1)\sigma^2)$ distribution. Thus, RVs $U_t$ will have (an approximately) log-normal distribution, shown with the solid line in Figure 8a. Note that this result is close to that obtained in [29]. Nevertheless, the distribution of the number of vaccinated population (Series B), shown with the solid line in Figure 8b, has a more pronounced "peak" close to the origin. It can also be explained by previous theoretical results, primarily given in Theorem 2, i.e., by Equation (8), which concerns the asymptotic behavior of the main GSB series $(y_t)$.

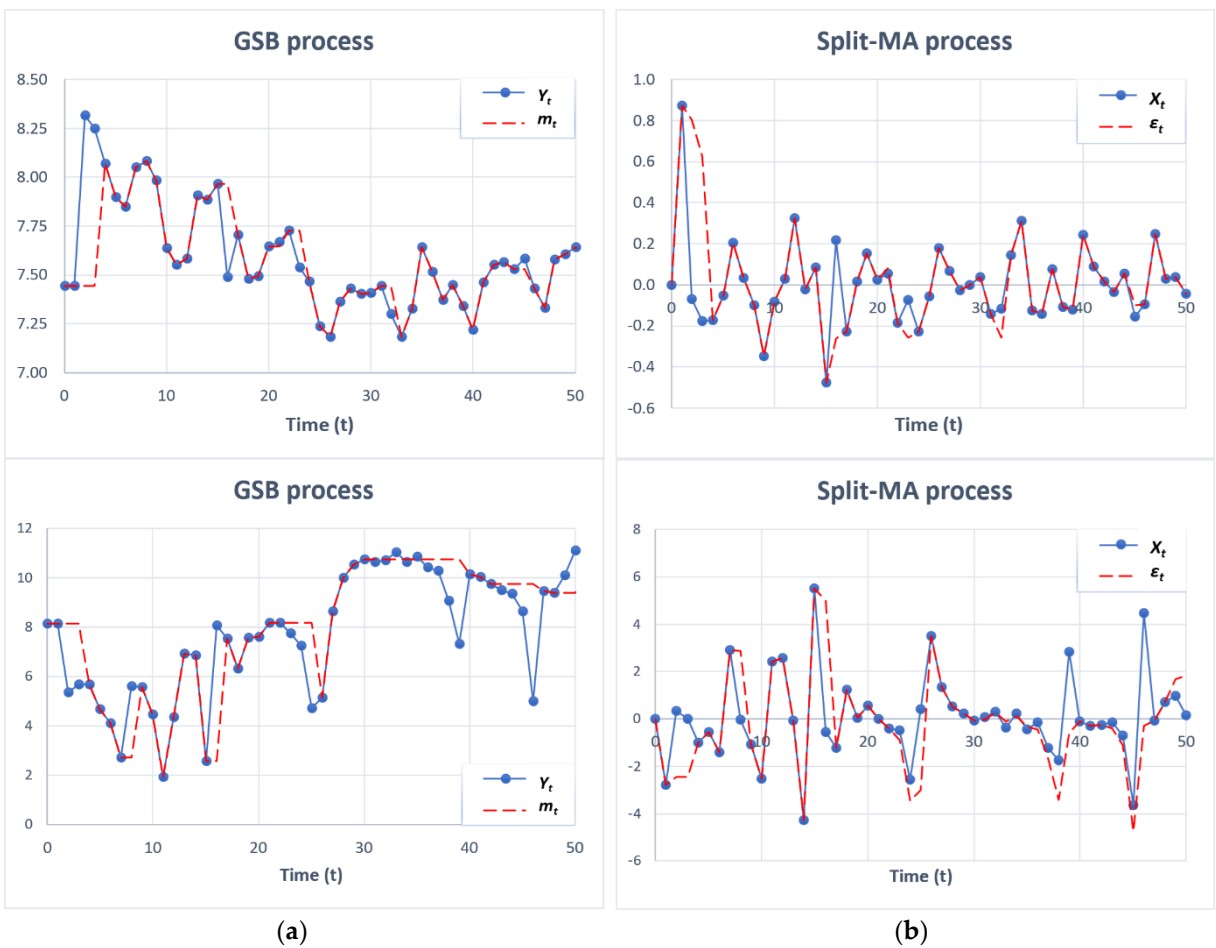

**Figure 7.** Graphs of empirical and modeled data: (**a**) log-volumes (solid lines) and martingale means (dashed lines); (**b**) Split-MA(1) process (solid lines) and innovations series (dashed lines). The upper panels represent the dynamics of the COVID-19 infection (Series A), and the lower panels represent the dynamics of the vaccinated population (Series B).

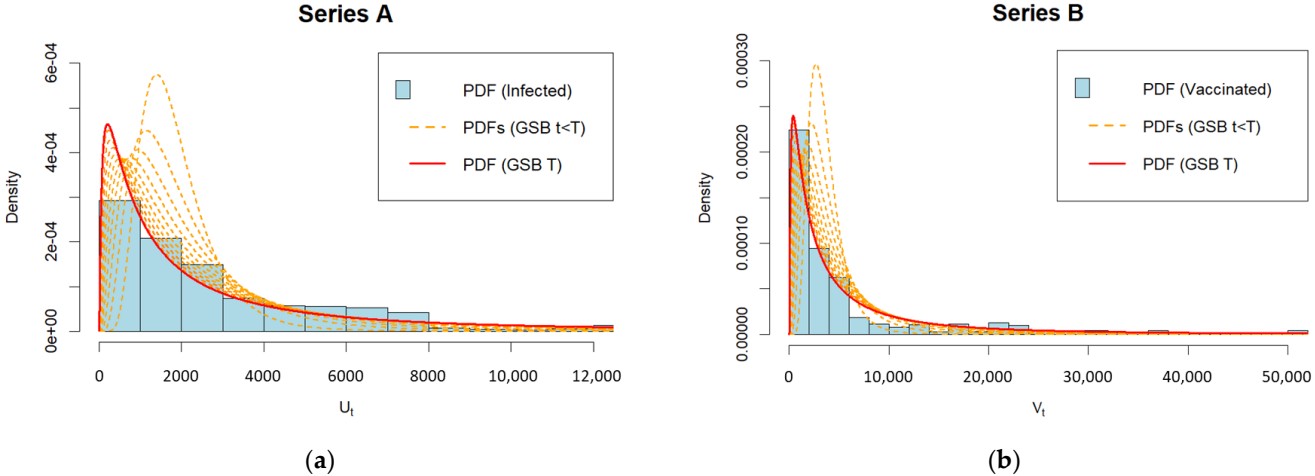

**Figure 8.** Empirical distributions of actual data (histograms) and their fitted PDFs (lines), obtained by the proposed estimation procedure: (**a**) distribution of the infected population (Series A); (**b**) distribution of the vaccinated population (Series B).

## 6. Conclusions

The stochastic analysis of the GSB process presented in this paper confirms its possibility in modeling actual time series with pronounced fluctuations. The applied methods of dynamic and statistical analysis, based on this process, aim here to understand the long-term tendency of the SARS-COV2 virus behavior, as well as the immunization process. Along with other contemporary research, we hope this one can help further development of successful methods of overcoming the pandemic. To this end, notice that new strains of the SARS-CoV2 virus, which are very common, can affect the overall symptoms as well as the disease dynamics of COVID-19 (see, c.f. [43–45]). They may therefore change the dynamics of both time series investigated here. This may therefore be a new goal and motivation for some future research.

Finally, let us emphasize that one of the main stochastic advantages of the GSB model is that it allows the simultaneous use of both stationary and non-stationary components. Thereby, the asymptotic behavior of the GSB time series as well as the corresponding estimates thus obtained are of particular importance. It should also be noted that the proposed parameter estimation procedure can be implemented algorithmically in a relatively simple way. Additionally, some other estimation methods, such as the Empirical Characteristic Function (ECF) method described in [12] can be used. As shown in [11,12], it can also be used to model some other types of real data with pronounced and persistent fluctuations.

**Author Contributions:** Conceptualization, M.J.; data curation, M.J.; formal analysis, V.S.; methodology, K.K.; project administration, B.P.; software, K.K. and B.P.; supervision, V.S.; validation, P.Č.; visualization, P.Č.; writing—original draft, M.J., V.S. and K.K.; writing—review and editing, B.P. All authors have read and agreed to the published version of the manuscript.

**Funding:** This research was funded by the Ministry of Education, Science and Technological Development of the Republic of Serbia. (Grant number: III 47016.)

**Data Availability Statement:** Not applicable.

**Acknowledgments:** The authors would like to thank the Electronic Government of the Republic of Serbia and the Institute for Public Health "Milan Jovanović-Batut" for providing datasets used in this research.

**Conflicts of Interest:** The authors declare no conflict of interest.

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
