# Peer review of "Asymptotic Properties and Application of GSB Process: A Case Study of the COVID-19 Dynamics in Serbia"

_mathematics, doi:10.3390/math10203849_

Round 1

Reviewer 1 Report

This paper discusses in depth a particular stochastic model (the GSB process) which offers certain advantages (in principle) over more conventional models, namely elements of non-linearity and non-stationarity. This is, again in principle, of potential advantage in situations where the fluctuations are pronounced. Previously, the authors have described aspects of the underlying theory which are pulled together in this paper. In addition, detailed consideration is given to the challenging issue of parameter estimation. Finally, an illustration is given in the context of analysing Covid-19 dynamics from the population in Serbia. The work is interesting, well presented, and it is good to see application to real data. However, before publication I feel the presentation can be improved by being more explicit in the discussion about why the results of the present study are superior to those obtained using more conventional models, or if they are not, where the shortcomings lie. To be useful and more widely adopted, it would be very helpful to have this spelled out. Moreover, it would be useful to have a very clear statement as to just how robust the parameter estimation procedures are - there are a lot of parameters which can sometimes be a cause for concern. Finally, a minor point, could the meaning of the totals given in Table 3 please be clarified?

Author Response

Authors’ Responses to Reviewer 1

  • The work is interesting, well presented, and it is good to see application to real data.

Authors’ response: We appreciate your comments as well as constructive suggestions for improving our manuscript. In this revised version, all suggestions have been considered and all questions have been answered point by point. (Please, see the parts of manuscript highlighted with red.)

  • However, before publication I feel the presentation can be improved by being more explicit in the discussion about why the results of the present study are superior to those obtained using more conventional models, or if they are not, where the shortcomings lie. To be useful and more widely adopted, it would be very helpful to have this spelled out.

Authors’ response: Thank you for this comment. In Introduction and new Subsection 5.2 we tried to describe in more detail our approach and compare it with some related models dealing with COVID-19 dynamics. (Please, see Lines 44-60 and Line 775 in the revised version). Some discussion about the results of the present study is also given in Conclusion. (Please, see Lines 783-786 in the revised version.)

  • Moreover, it would be useful to have a very clear statement as to just how robust the parameter estimation procedures are - there are a lot of parameters which can sometimes be a cause for concern.

Authors’ response: Thank you very much for this comment. We fully agree with your statement. Although the GSB process has (only) three unknown parameters, their estimates can be obtained in several ways, and we tried to describe all those possibilities. Based on your suggestion, we have highlighted the most robust estimates used in modeling real-world COVID-19 data. (Please, see Lines 663-666 and Lines 721-723 in the revised version.)

  • Finally, a minor point, could the meaning of the totals given in Table 3 please be clarified?

Authors’ response: Thank you for this comment. These values represent total sums of the corresponding data. However, considering that they are not of great importance, they are omitted in this version of the manuscript.

Reviewer 2 Report

The study in general seems interesting. The authors should present clearly their objectives in the Introduction. I think the authors should also give some indication on the main features of the disease.  Please, comment the advantages of this methodology comparatively to alternative ones on this subject.

A section on the limitations of the study and eventually possible alternative methodologies to study the phenomenon could be introduced, to give other insights on the subject under study.

The results shall be complemented to better adjust to the results.

In the abstact, in line 11 delete "named".

Please ask an English expert to review and make necessary adustments in the text.

Author Response

Authors’ Responses to Reviewer 2

  • The study in general seems interesting.

Authors’ response: We appreciate your comments as well as suggestions for improving our manuscript. In this revised version, all suggestions have been considered and all questions have been answered point by point. (Please, see the parts of manuscript highlighted with red.)

  • The authors should clearly present their objectives in the Introduction. I think the authors should also give some indication on the main features of the disease. Please, comment the advantages of this methodology comparatively to alternative ones on this subject.

Authors’ response: Thank you for your comment. In the Introduction section, we stated: 'The main goal of this paper is a more detailed investigation of the non-stationary components (time series) of the GSB model.' It was defined from a mathematical point of view and in accordance with the goals of the "Mathematics" journal. Reading Your comments, we realize that it might not be so clear that our goal was not the analysis of the indicators of the COVID-19 disease nor to explain its main features. Our goal was to formally study the stochastic-mathematical model, which can be applied, among others, in modeling the time-dynamics of this disease. Therefore, based on your suggestions, we added the paragraph in the Introduction. (Please, see Lines 44-60 in the revised version.)

  • A section on the limitations of the study and eventually possible alternative methodologies to study the phenomenon could be introduced, to give other insights on the subject under study.

Authors’ response: Thank you for your comment. Following your suggestions, in Introduction and newly added Subsection 5.2 we described the methodology of applying our model and compared it with some similar alternative approaches to this topic.

  • In the abstract, in line 11 delete "named".

Authors’ response: Thank you, it has now been corrected.

  • Please ask an English expert to review and make necessary adjustments in the text.

Authors’ response: Thank you. We have given our manuscript to a native English-speaking colleague for a thorough check. According to his suggestions, appropriate changes were made to the text.

Reviewer 3 Report

Please find attached the review report. 

Author Response

Authors’ Responses to Reviewer 3

  • The article contributes to stochastic modelling. It is well written and fits within the scope of Mathematics. However, in my opinion, it requires a minor revision.

Authors’ response: We appreciate your comments as well as constructive suggestions for improving our manuscript. In this revised version, all suggestions have been considered and all questions have been answered point by point. (Please, see the parts of manuscript highlighted with red.)

  • Line 122: Please justify the convergences.

Authors’ response: Thank you for this comment. Both convergences are now described and shown in more detail. (Please, see Lines 142 and 143 in the revised version.)

  • Line 170: The variance of 2ϵt is equal to 4σ 2

Authors’ response: Thank you very much for this comment. This typo has now been corrected. (Please, see Lines 194 and 195 in the revised version.)

  • Lines 302–304: Please justify the formula in more details.

Authors’ response: Thank you for this comment. This formula is now derived in detail. (Please, see Lines 324-330 in the revised version.)

  • Please outline further research plans in Conclusion.

Authors’ response: Thank you very much for this comment. The conclusion has now been changed and supplemented based on your suggestion. (Please, see Lines 782-786 and Lines 792-795 in the revised version.)

Reviewer 4 Report

The manuscript studies a Gaussian (or Generalized) Split-BREAK (GSB) process. Under the normality assumptions, the authors consider parameter estimation and asymptotic properties of the derived estimators. The paper provides results of numerical simulations and applies the GSB model to analyse the dynamics of infected and immunized population against the disease COVID19 in the Republic of Serbia. The topic is a very interesting that deserves attention. The paper appears to be well-referenced. I have only minor comments on the manuscript.

Line 12. The abbreviation GSB is only introduced in lines 33-34. It would be good to explain the abbreviation GSB in the abstract as well.

Lines 56 and 125. Both y_t (lower case) and X_t (upper case) are random variables. Shouldn’t both of them be upper case?

Line 61. It may be a good idea to mention here that E and V are the expectation (or expected value) and the variance, respectively. What probability measure is used?

Line 65. Some readers may not be familiar with the “as” (almost sure) notation. It would be good to explain it here (see convergence in distribution in lines 214-216). Also, check the font: in Line 65, “as” is in italic, while it is in roman in Line 511, for example.

Line 130 (see also Line 200). Is the “I” the indicator function? Introduce the notation.

Line 192 (see also lines 232 and 236). Insert a reference to Levy’s theorem.

Lines 211 and 212. The variable “j” in the convolutions is from 1 to t but there is no “j” in the expressions on the right-hand side of formulas (6) and (7).

Line 217. It is better to use “let us” instead of “let’s”.

Lines 361, 579 and 677. Change “-0,5” to “0.5”.

Figures 4 and 5. The figures look a bit blurry. It would be good to use eps- and pdf-files if you do not use them already.

Figures 4, 5, 6 and 8. Consider changing the layout: from 2 plots in one row to 2 plots in one column.

Line 529. Insert a reference to Hoeffding-Robbins theorem.

Line 563. Consider changing “min” to “\min”.

Author Response

Authors’ Responses to Reviewer 4

  • The topic is a very interesting that deserves attention. The paper appears to be well-referenced. I have only minor comments on the manuscript.

Authors’ response: We appreciate your comments as well as constructive and detailed suggestions for improving our manuscript. In this revised version, all suggestions have been considered and all questions have been answered point by point. (Please, see the parts of manuscript highlighted with red.)

  • Line 12. The abbreviation GSB is only introduced in lines 33-34. It would be good to explain the abbreviation GSB in the abstract as well.

Authors’ response: Thank you. It has now been corrected.

  • Lines 56 and 125. Both y_t (lower case) and X_t (upper case) are random variables. Shouldn’t both be upper case?

Authors’ response: Thank you for this comment. We agree with you that random variables (RVs) are usually capitalized. Our idea was to label the non-stationary components (y_t) and (m_t) with lowercase letters, as opposed to the stationary one (X_t). This notation was used in the original work on the STOPBREAK process (Engle & Smith, 1998). However, if the Reviewer insists, we are ready to make such a correction.

  • Line 61. It may be a good idea to mention here that E and V are the expectation (or expected value) and the variance, respectively. What probability measure is used?

Authors’ response: Thank you very much for this comment. The labels for expectation and variance, as well as the probability space on which they are observed, are now explained in more detail. (Please, see Lines 76-81 in the revised version.)

  • Line 65. Some readers may not be familiar with the “as” (almost sure) notation. It would be good to explain it here (see convergence in distribution in lines 214-216). Also, check the font: in Line 65, “as” is in italic, while it is in roman in Line 511, for example.

Authors’ response: Thank you very much for this comment. It has now been corrected. (Please, see Line 86, as well as other "as" notations in the revised version.)

  • Line 130 (see also Line 200). Is the “I” the indicator function? Introduce the notation.

Authors’ response: Thank you. The indicator function I(∙) was already defined before when the RV q_t (Noise Indicator) was introduced. (Please, see Line 89 in the revised version.)

  • Line 192 (see also lines 232 and 236). Insert a reference to Levy’s theorem.

Authors’ response: Thank you. It has now been corrected.

  • Lines 211 and 212. The variable “j” in the convolutions is from 1 to t but there is no “j” in the expressions on the right-hand side of formulas (6) and (7).

Authors’ response: Thank you for this comment. Indeed, the right-hand sides in Eqs. (6) and (7) contain CDFs with the same distributions. Now, we have added the "j" subscripts and (hopefully) adequately displayed and explained these convolutions.

  • Line 217. It is better to use “let us” instead of “let’s”.

Authors’ response: Thank you. It has now been corrected.

  • Lines 361, 579 and 677. Change “-0,5” to “0.5”.

Authors’ response: Thank you. The value "-0.5", i.e., the condition is necessary for the probability  to be within (0,1) interval. (Please, see Eq.(14) in Line 382, as well as Lines 388-390 in the revised version.)

  • Figures 4 and 5. The figures look a bit blurry. It would be good to use eps- and pdf-files if you do not use them already.

Authors’ response: Thank you for this comment. These Figures were created using the software "Wolfram Mathematica", while the manuscript was written in the text-processor "MS Word". This may be the cause of the blurring of them. We have now corrected and reduced the size of these Figures, so hopefully they are now clearer.

  • Figures 4, 5, 6 and 8. Consider changing the layout: from 2 plots in one row to 2 plots in one column.

Authors’ response: Thank you for this comment. In accordance with the journal's template, we feel that the way of displaying Figures that we have used here is appropriate. First, because it allows the referencing of left image (a) and right image (b).

  • Line 529. Insert a reference to Hoeffding-Robbins theorem.

Authors’ response: Thank you. It has now been corrected.

  • Line 563. Consider changing “min” to “\min”.

Authors’ response: Thank you. It has now been corrected.

Round 2

Reviewer 2 Report

The paper has been improved and it is now better than before. Authors made a good exercise on contextualizing the problem. I suggest anyway that the authors present the limitations of the method on this problem. Are all the previous conditions satisfied for the application of the method to these data? After that the paper will need the formatting and a final check to make last adjustments and checks before the publication.

Author Response

Responses to Reviewer 2

  • The paper has been improved and it is now better than before. Authors made a good exercise on contextualizing the problem.

Authors’ response: Thank you. We appreciate your comments as well as suggestions for improving our manuscript. In this revised version, all suggestions have been considered and all questions have been answered point by point. (Please, see the parts of manuscript highlighted with red.)

  • I suggest anyway that the authors present the limitations of the method on this problem. Are all the previous conditions satisfied for the application of the method to these data?

Authors’ response:  Thank you. Following your suggestions, in this version we have added some more comments in the Introduction (please, see Lines 60-65), as well as in the Conclusion (please, see Lines 793-800). Also, some more current references have been added (please, see Lines 874-880 and Lines 894-898). Finally, we would like to point out that, in our opinion, the introductory part and the newly added Subsec. 5.2 provided sufficient arguments that justify the application of our model. For additional clarification, some parts of Subsec. 5.2 are also modified in this version (please, see Lines 673-689).